# Which Experiences Are Influential for RL Agents? Efficiently Estimating The Influence of Experiences

## Abstract

In reinforcement learning (RL) with experience replay, experiences stored in a replay buffer influence the RL agent's performance. Information about how these experiences influence the agent's performance is valuable for various purposes, such as identifying experiences that negatively influence underperforming agents. One method for estimating the influence of experiences is the leave-one-out (LOO) method. However, this method is usually computationally prohibitive. In this paper, we present Policy Iteration with Turn-over Dropout (PIToD), which efficiently estimates the influence of experiences. We evaluate how accurately PIToD estimates the influence of experiences and its efficiency compared to LOO. We then apply PIToD to amend underperforming RL agents, i.e., we use PIToD to estimate negatively influential experiences for the RL agents and to delete the influence of these experiences. We show that RL agents' performance is significantly improved via amendments with PIToD.

## 1 Introduction

In reinforcement learning (RL) with experience replay, the performance of an RL agent is influenced by experiences. Experience replay (Lin, 1992) is a data-generation mechanism indispensable in modern off-policy RL methods (Mnih et al., 2015; Hessel et al., 2018; Haarnoja et al., 2018a; Kumar et al., 2020). It allows an RL agent to learn from past experiences. These experiences influence the RL agent's performance (e.g., cumulative rewards) (Fedus et al., 2020). Estimating how each experience influences the RL agent's performance could provide useful information for many purposes. For example, we could improve the RL agent's performance by identifying and deleting negatively influential experiences. The capability to estimate the influence of experience will be crucial, as RL is increasingly applied to tasks where agents must learn from experiences of diverse quality (e.g., a mixture of experiences from both expert and random policies) (Fu et al., 2020; Yu et al., 2020; Agarwal et al., 2022; Smith et al., 2023; Liu et al., 2024; Tirumala et al., 2024).

However, estimating the influence of experiences with feasible computational cost is not trivial. One might consider estimating it by a leave-one-out (LOO) method (left part of Figure 1), which retrains an RL agent for each possible experience deletion. As we will discuss in Section 3, this method has quadratic time complexity and quickly becomes intractable due to the necessity of retraining.

In this paper, we present PIToD, a policy iteration (PI) method that efficiently estimates the influence of experiences (right part of Figure 1). PI is a fundamental method for many RL methods (Section 2). PIToD is PI augmented with turn-over dropout (ToD) (Kobayashi et al., 2020) to efficiently estimate the influence of experiences without retraining an RL agent (Section 4). We evaluate how accurately PIToD estimates the influence of experiences and its efficiency compared to the LOO method (Section 5). We then apply PIToD to amend underperforming RL agents by identifying and deleting negatively influential experiences (Section 6). To our knowledge, our work is the first to: (i) estimate the influence of experiences on the performance of RL agents with feasible computational cost, and (ii) modify RL agents' performance simply by deleting influential experiences.

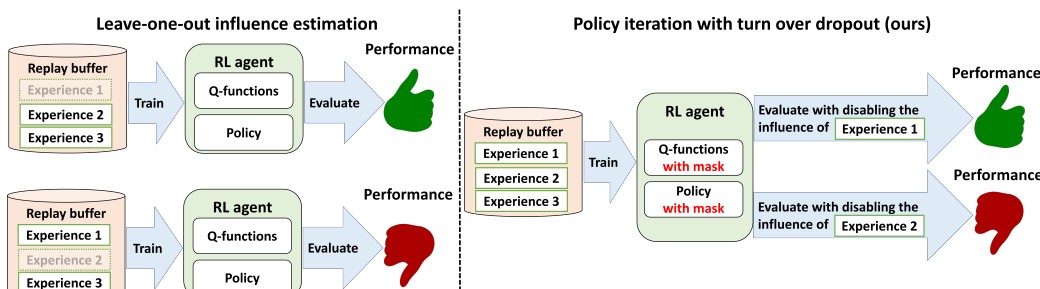

Figure 1: Leave-one-out (LOO) influence estimation method (left part) and our method (right part). LOO estimates the influence of experiences by retraining an RL agent for each experience deletion. In contrast, our method estimates the influence of experiences without retraining.

## 2 PRELIMINARIES

In Section 4, we will introduce our PI method for estimating the influence of experiences in the RL problem. As preliminaries for this, we explain the RL problem, PI, and influence estimation.

**Reinforcement learning (RL).** RL addresses the problem of an agent learning to act in an environment. The environment provides the agent with a state $s$. The agent responds by selecting an action $a$, and then the environment provides a reward $r$ and the next state $s'$. This interaction between the agent and environment continues until the agent reaches a terminal state. The agent aims to find a policy $\pi : \mathcal{S} \times \mathcal{A} \to [0, 1]$ that maximizes cumulative rewards (return). A Q-function $Q : \mathcal{S} \times \mathcal{A} \to \mathbb{R}$ is used to estimate the expected return.

**Policy iteration (PI).** PI is a method for solving RL problems. PI updates the policy and Q-function by iteratively performing policy evaluation and improvement. Many implementations of policy evaluation and improvement have been proposed (e.g., Lillicrap et al. (2015); Fujimoto et al. (2018); Haarnoja et al. (2018a)). In the main part of this paper, we focus on the policy evaluation and improvement used in Deep Deterministic Policy Gradient (DDPG). In policy evaluation in DDPG, the Q-function $Q_\phi : \mathcal{S} \times \mathcal{A} \to \mathbb{R}$, parameterized by $\phi$, is updated as:

$$\phi \leftarrow \phi - \nabla_\phi \mathbb{E}_{(s,a,r,s') \sim \mathcal{B}, \ a' \sim \pi_\theta(\cdot|s')} \left[ \left( r + \gamma Q_{\bar{\phi}}(s', a') - Q_\phi(s, a) \right)^2 \right], \tag{1}$$

where $\mathcal{B}$ is a replay buffer containing the collected experiences, and $Q_{\bar{\phi}}$ is a target Q-function. In policy improvement in DDPG, policy $\pi_\theta$, parameterized by $\theta$, is updated as:

$$\theta \leftarrow \theta + \nabla_\theta \mathbb{E}_{s \sim \mathcal{B}, \ a_\theta \sim \pi_\theta(\cdot|s)} \left[ Q_\phi(s, a_\theta) \right]. \tag{2}$$

**Estimating the influence of experiences.** Given the policy and Q-functions updated through PI, we aim to estimate the influence of experiences on performance. Formally, letting $e_i$ be the $i$-th experience contained in the replay buffer $\mathcal{B}$, we evaluate the influence of $e_i$ as

$$L \left( Q_{\phi, \mathcal{B} \setminus \{e_i\}}, \pi_{\theta, \mathcal{B} \setminus \{e_i\}} \right) - L \left( Q_{\phi, \mathcal{B}}, \pi_{\theta, \mathcal{B}} \right), \tag{3}$$

where $L$ is a metric for evaluating the performance of the Q-function and policy, $Q_{\phi, \mathcal{B}}$ and $\pi_{\theta, \mathcal{B}}$ are the Q-function and policy updated with all experiences contained in $\mathcal{B}$, and $Q_{\phi, \mathcal{B} \setminus \{e_i\}}$ and $\pi_{\theta, \mathcal{B} \setminus \{e_i\}}$ are the ones updated with $\mathcal{B}$ other than $e_i$. $L$ is defined according to the focus of the experiments. In this paper, we define $L$ as policy and Q-function loss for the experiments in Section 5, and as empirical return and Q-estimation bias for the applications in Section 6.

## 3 LEAVE-ONE-OUT (LOO) INFLUENCE ESTIMATION

What method can be used to estimate the influence of experiences? One straightforward method is based on the LOO algorithm (Algorithm 1). This algorithm estimates the influence of experiences

---

**Algorithm 1** Leave-one-out influence estimation for policy iteration

---
1: **given** replay buffer $\mathcal{B}$, learned parameters $\phi, \theta$, and number of policy iteration $I$.
2: **for** $e_i \in \mathcal{B}$ **do**
3:   Initialize temporal parameters $\phi'$ and $\theta'$.
4:   **for** $I$ iterations **do**
5:     Update $Q_{\phi'}$ with $\mathcal{B} \backslash \{e_i\}$ (policy evaluation).
6:     Update $\pi_{\theta'}$ with $\mathcal{B} \backslash \{e_i\}$ (policy improvement).
7:   Evaluate the influence of $e_i$ as

$$L\left(Q_{\phi'}, \pi_{\theta'}\right) - L\left(Q_{\phi}, \pi_{\theta}\right). \tag{4}$$

---

by retraining the RL agent's components (i.e., policy and Q-functions) for each experience deletion. Specifically, it retrains the policy $\pi_{\theta'}$ and Q-function $Q_{\phi'}$ using $\mathcal{B} \backslash \{e_i\}$ through $I$ policy iterations (lines 4–6). Here, $I$ equals the number of policy iterations required for training the original policy $\pi_{\theta}$ and Q-function $Q_{\phi}$. After retraining the components, the influence of $e_i$ is evaluated using Eq. 4 with $\pi_{\theta'}, Q_{\phi'}$ and $\pi_{\theta}, Q_{\phi}$ (line 7).

However, in typical settings, Algorithm 1 becomes computationally prohibitive due to retraining. In typical settings (e.g., Fujimoto et al. (2018); Haarnoja et al. (2018b)), the size of the buffer $\mathcal{B}$ is small at the beginning of policy iteration and increases by one with each iteration. Consequently, the size of $\mathcal{B}$ is approximately equal to the number of iterations $I$ (i.e., $|\mathcal{B}| \approx I$). Since Algorithm 1 retrains the RL agent's components through $I$ policy iterations for each $e_i$, the total number of policy iterations across the entire algorithm becomes $I^2$. The value of $I$ typically ranges between $10^3$ and $10^6$ (e.g., Chen et al. (2021a); Haarnoja et al. (2018b)), which makes it difficult to complete all policy iterations in a realistic timeframe.

In the next section, we will introduce a method to estimate the influence of experiences without retraining the RL agent's components.

## 4 POLICY ITERATION WITH TURN-OVER DROPOUT (PITOD)

In this section, we present **P**olicy **I**teration with **T**urn-**o**ver **D**ropout (PIToD), which estimates the influence of experiences without retraining. The concept of PIToD is shown in Figure 2, and an algorithmic description of PIToD is shown in Algorithm 2. Inspired by ToD (Kobayashi et al., 2020), PIToD uses masks and flipped masks to drop out the parameters of the policy and Q-function. Further details are provided in the following paragraphs.

**Masks and flipped masks.** PIToD uses mask $\mathbf{m}_i$ and flipped mask $\mathbf{w}_i$, which are binary vectors uniquely associated with experience $e_i$. The mask $\mathbf{m}_i$ consists of elements randomly initialized to 0 or 1. $\mathbf{m}_i$ is used to drop out the parameters of the policy and Q-function during PI with $e_i$. Additionally, the flipped mask $\mathbf{w}_i$ is the negation of $\mathbf{m}_i$, i.e., $\mathbf{w}_i = \mathbf{1} - \mathbf{m}_i$. $\mathbf{w}_i$ is used to drop out the parameters of the policy and Q-function for estimating the influence of $e_i$.

**Policy iteration with the mask (lines 5–6 in Algorithm 2).** PIToD applies $\mathbf{m}_i$ to the policy and Q-function during PI with $e_i$. It executes PI with variants of policy evaluation (Eq. 1) and improvement (Eq. 2) where masks are applied to the parameters of the policy and Q-function. The policy evaluation for PIToD is

$$\phi \leftarrow \phi - \nabla_{\phi} \mathbb{E}_{e_i=(s,a,r,s',i)\sim\mathcal{B},\ a'\sim\pi_{\theta,\mathbf{m}_i}(\cdot|s')} \left[ \left( r + \gamma Q_{\bar{\phi},\mathbf{m}_i}(s',a') - Q_{\phi,\mathbf{m}_i}(s,a) \right)^2 \right]. \tag{5}$$

The policy improvement for PIToD is

$$\theta \leftarrow \theta + \nabla_{\theta} \mathbb{E}_{e_i=(s,i)\sim\mathcal{B},\ a_{\theta,\mathbf{m}_i}\sim\pi_{\theta,\mathbf{m}_i}(\cdot|s)} \left[ Q_{\phi,\mathbf{m}_i}(s, a_{\theta,\mathbf{m}_i}) \right]. \tag{6}$$

Here, $Q_{\phi,\mathbf{m}_i}$ and $\pi_{\theta,\mathbf{m}_i}$ are the Q-function and policy to which the mask $\mathbf{m}_i$ is applied. In Eq. 5 and Eq. 6, for inputs from $e_i$, $Q_{\phi,\mathbf{m}_i}$ and $\pi_{\theta,\mathbf{m}_i}$ compute their outputs without using the parameters that are dropped out by $\mathbf{m}_i$. Thus, the parameters dropped out by $\mathbf{m}_i$ (i.e., the parameters obtained by applying $\mathbf{w}_i$) are expected not to be influenced by $e_i$. More theoretically, if $Q_{\phi,\mathbf{m}_i}$ and $\pi_{\theta,\mathbf{m}_i}$

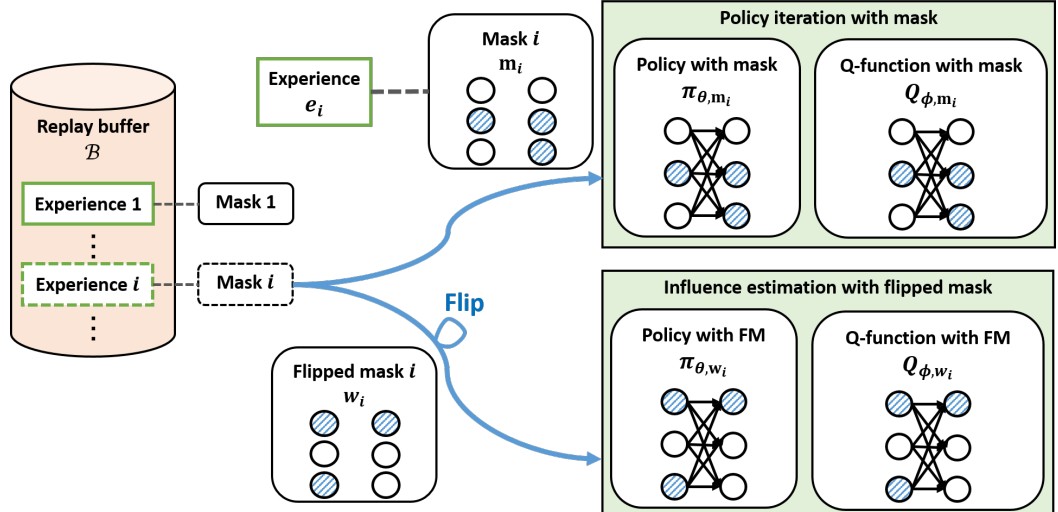

Figure 2: The concept of PIToD. PIToD uses mask $\mathbf{m}_i$ and flipped mask $\mathbf{w}_i$. It applies $\mathbf{m}_i$ to the policy and Q-function for PI with $e_i$. Additionally, it applies $\mathbf{w}_i$ to the policy and Q-function for estimating the influence of $e_i$.

---

**Algorithm 2** Policy iteration with turn-over dropout (PIToD)

---

1: Initialize policy parameters $\theta$, Q-function parameters $\phi$, and an empty replay buffer $\mathcal{B}$; Set influence estimation interval $I_{\text{ie}}$.
2: **for** $i' = 0, ..., I$ iterations **do**
3:     Take action $a \sim \pi_\theta(\cdot|s)$; Observe reward $r$ and next state $s'$. Define an experience using $i'$ as: $e_{i'} = (s, a, r, s', i')$; $\mathcal{B} \leftarrow \mathcal{B} \bigcup \{e_{i'}\}$.
4:     Sample experiences $\{(s, a, r, s', i), ...\}$ from $\mathcal{B}$ (Here, $e_i = (s, a, r, s', i)$).
5:     Update $\phi$ with gradient descent using
$$\nabla_\phi \sum_{(s,a,r,s',i)} \left( r + \gamma Q_{\bar{\phi},\mathbf{m}_i}(s', a') - Q_{\phi,\mathbf{m}_i}(s,a) \right)^2, \;\; a' \sim \pi_{\theta,\mathbf{m}_i}(\cdot|s').$$
6:     Update $\theta$ with gradient ascent using
$$\nabla_\theta \sum_{(s,i)} Q_{\phi,\mathbf{m}_i}(s, a_{\theta,\mathbf{m}_i}), \;\; a_{\theta,\mathbf{m}_i} \sim \pi_{\theta,\mathbf{m}_i}(\cdot|s).$$
7:     **if** $i'\%I_{\text{ie}} = 0$ **then**
8:         For $e_i \in \mathcal{B}$, estimate the influence of $e_i$ using
$$L\left(Q_{\phi,\mathbf{w}_i}, \pi_{\theta,\mathbf{w}_i}\right) - L\left(Q_\phi, \pi_\theta\right) \;\; \text{or} \;\; L\left(Q_{\phi,\mathbf{w}_i}, \pi_{\theta,\mathbf{w}_i}\right) - L\left(Q_{\phi,\mathbf{m}_i}, \pi_{\theta,\mathbf{m}_i}\right).$$

---

are dominantly influenced by $e_i$, the parameters obtained by $\mathbf{w}_i$ are provably not influenced by $e_i$ (see Appendix A for details). Based on this theoretical property, we estimate the influence of $e_i$ by applying $\mathbf{w}_i$ to policy and Q-functions (see the next paragraph for details).

**Estimating the influence of experience with flipped mask (lines 7–8 in Algorithm 2).** PIToD periodically estimates the influence of $e_i$ by applying $\mathbf{w}_i$ to the policy and Q-function. It estimates the influence of $e_i$ (Eq. 3) as

$$L\left(Q_{\phi,\mathbf{w}_i}, \pi_{\theta,\mathbf{w}_i}\right) - L\left(Q_\phi, \pi_\theta\right), \tag{7}$$

where the first term is the performance when $e_i$ is deleted, and the second term is the performance with all experiences. $Q_{\phi,\mathbf{w}_i}$ and $\pi_{\theta,\mathbf{w}_i}$ are the Q-function and policy with dropout based on $\mathbf{w}_i$. $Q_\phi$ and $\pi_\theta$ are the Q-function and policy without dropout. For the second term, if we want to highlight the influence of $e_i$ more significantly, the term can be evaluated by alternatively using the masked policy and Q-functions: $L\left(Q_{\phi,\mathbf{m}_i}, \pi_{\theta,\mathbf{m}_i}\right)$. The influence estimation is performed every $I_{\text{ie}}$

iterations (line 7 in Algorithm 2). These influence estimations by PIToD do not require retraining for each experience deletion, unlike the LOO method.

**Implementation details for PIToD.** For the experiments in Sections 5 and 6, each mask element is initialized to 0 or 1, drawn from a discrete uniform distribution, to minimize overlap between the masks (see Appendix B for details). Additionally, we implemented PIToD using Soft Actor-Critic (Haarnoja et al., 2018b) for these experiments (see Appendix C for details).

## 5 EVALUATIONS FOR PIToD

In the previous section, we introduced PIToD, a method that efficiently estimates the influence of experiences. In this section, we evaluate its accuracy in influence estimation (Section 5.1) and its computational efficiency (Section 5.2).

### 5.1 HOW ACCURATELY DOES PIToD ESTIMATE THE INFLUENCE OF EXPERIENCES? EVALUATIONS WITH SELF-INFLUENCE

In this section, we evaluate how accurately PIToD estimates the influence of experiences by focusing on their self-influence. Self-influence is the influence of an experience on prediction performance using that same experience. We define self-influences on policy evaluation and on policy improvement. The self-influence of an experience $e_i := (s, a, r, s', i)$ on policy evaluation is

$$L_{pe,i}(Q_{\phi, \mathbf{w}_i}) - L_{pe,i}(Q_{\phi, \mathbf{m}_i}), \tag{8}$$

$$\text{where } L_{pe,i}(Q) = \left(r + \gamma Q_{\bar{\phi}, \mathbf{m}_i}(s', a') - Q(s, a)\right)^2, \ a' \sim \pi_{\theta, \mathbf{m}_i}(\cdot|s').$$

Here, $L_{pe,i}$ represents the temporal difference error based on $e_i$. The self-influence of $e_i$ on policy improvement is

$$L_{pi,i}(\pi_{\theta, \mathbf{w}_i}) - L_{pi,i}(\pi_{\theta, \mathbf{m}_i}), \text{ where } L_{pi,i}(\pi) = Q_{\phi, \mathbf{m}_i}(s, a'), \quad a' \sim \pi(\cdot|s). \tag{9}$$

Here, $L_{pi,i}$ represents the Q-value estimate based on $e_i$.

We evaluate whether PIToD has correctly estimated the influence of experiences by examining the signs (positive or negative) of the values of Eq. 8 and Eq. 9. If PIToD has correctly estimated the influence of experiences, the value of Eq. 8 should be positive. $Q_{\phi, \mathbf{m}_i}$ is optimized by PIToD to minimize $L_{pe,i}$ (line 5 in Algorithm 2), while $Q_{\phi, \mathbf{w}_i}$ is not. Therefore, $L_{pe,i}(Q_{\phi, \mathbf{m}_i}) \leq L_{pe,i}(Q_{\phi, \mathbf{w}_i})$, implying that Eq. 8 $\geq 0$. Conversely, if PIToD has correctly estimated, the value of Eq. 9 should be negative. $\pi_{\theta, \mathbf{m}_i}$ is optimized by PIToD to maximize $L_{pi,i}$ (line 6 in Algorithm 2), while $\pi_{\theta, \mathbf{w}_i}$ is not. Therefore, $L_{pi,i}(\pi_{\theta, \mathbf{m}_i}) \geq L_{pi,i}(\pi_{\theta, \mathbf{w}_i})$, which implies that Eq. 9 $\leq 0$.

We periodically evaluate the ratio of experiences for which PIToD has correctly estimated self-influence in the MuJoCo environments (Todorov et al., 2012). The MuJoCo tasks for this evaluation are Hopper, Walker2d, Ant, and Humanoid. In this evaluation, 5000 policy iterations (i.e., lines 3–6 of Algorithm 2) constitute one epoch, with 125 epochs allocated for Hopper and 300 epochs for the others. At each epoch, we (i) calculate the self-influence (Eq. 8 and Eq. 9) of experiences stored in the replay buffer and (ii) record the ratio of experiences for which PIToD has correctly estimated self-influence.

Evaluation results (Figure 3) show that the ratio of experiences for which the self-influence (Eq. 8 and Eq. 9) is correctly estimated exceeds the chance rate of 0.5. For self-influence on policy evaluation (Eq. 8), the ratio of correctly estimated experiences is higher than 0.9 across all environments. Furthermore, for self-influence on policy improvement (Eq. 9), the ratio of correctly estimated experiences exceeds 0.7 in Hopper, 0.8 in Walker2d and Ant, and 0.9 in Humanoid. These results suggest that PIToD estimates the influence of experiences more accurately than random estimation.

Figure 3 also shows that in policy improvement, the ratio of correctly estimated experiences tends to be higher in higher-dimensional environments (Hopper < Walker = Ant < Humanoid). This suggests that the policy tends to fit more significantly to each experience in higher-dimensional environments. Additionally, in both policy evaluation and improvement, the ratio gradually decreases as the epoch progresses. As the epoch progresses, the ratio of experiences to the policy and Q-network size increases. We hypothesize that this makes tracking the influence of each experience more difficult.

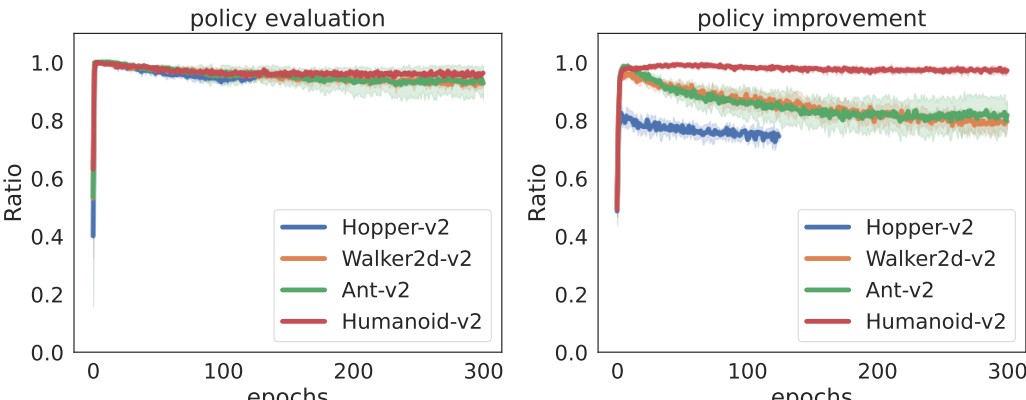

Figure 3: The ratio of experiences for which PIToD correctly estimated self-influence. The left-hand figure displays this ratio in policy evaluation cases, where a positive self-influence value (i.e., Eq. 8 $\geq$ 0) is correct. The right-hand figure displays the ratio in policy improvement cases, where a negative self-influence value (i.e., Eq. 9 $\leq$ 0) is correct. In both figures, the vertical axis represents the ratio of correctly estimated experiences, and the horizontal axis shows the number of epochs. In both cases, the ratio of correctly estimated experiences surpasses the chance rate of 0.5.

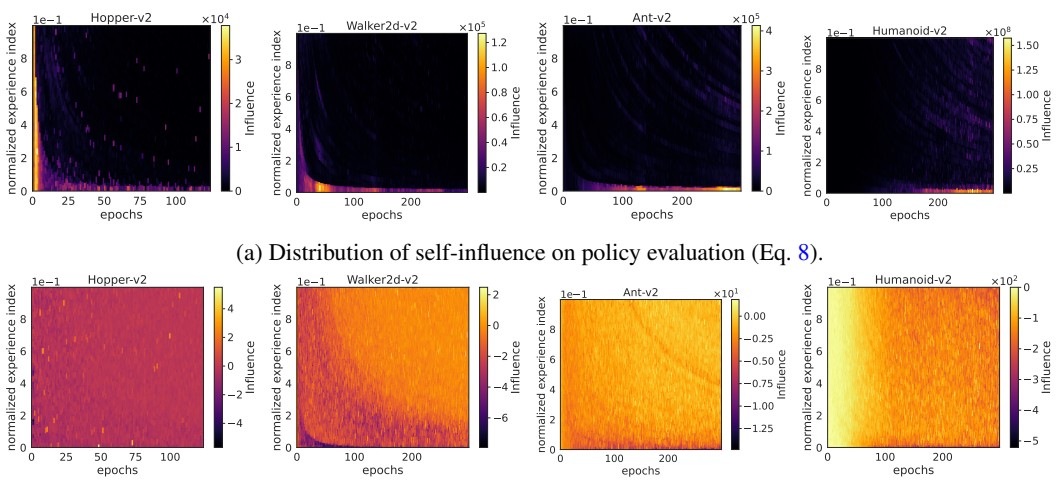

(a) Distribution of self-influence on policy evaluation (Eq. 8).

(b) Distribution of self-influence on policy improvement (Eq. 9).

Figure 4: Distribution of self-influence on policy evaluation and policy improvement. The vertical axis represents the normalized experience index, which ranges from 0.0 for the oldest experiences to 1.0 for the most recent experiences. This index corresponds to the normalized $i$ used in Algorithm 2. The horizontal axis represents the number of epochs. The color bar represents the value of self-influence. **Interpretation of this figure:** For example, if the value of self-influence for $e_i$ in policy evaluation cases is $2 \cdot 10^8$, this indicates that the value of $L_{pe,i}(Q_{\phi,\mathbf{w}_i})$ is $2 \cdot 10^8$ larger than that of $L_{pe,i}(Q_{\phi,\mathbf{m}_i})$. **Key insight:** In policy evaluation, experiences with high self-influence tend to concentrate on older ones (with smaller normalized experience indexes) as the epochs progress.

**Supplementary analysis.** How are experiences that exhibit significant self-influence distributed? Figure 4 shows the distribution of self-influence across experiences. From the figure, we see that in policy evaluation, the self-influence of older experiences (with smaller normalized experience indexes) becomes more significant as the epoch progresses. Conversely, for policy improvement, we observe no clear pattern in the distribution of influential experiences.

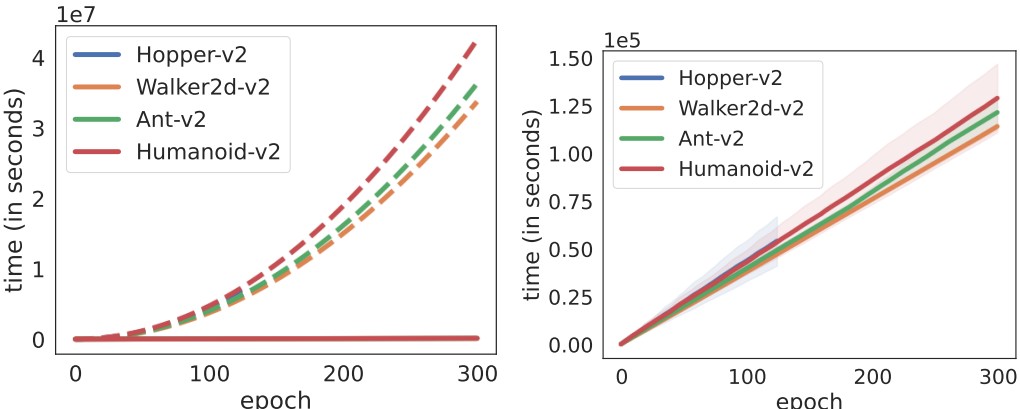

Figure 5: Wall-clock time required for influence estimation by PIToD and LOO. The solid line represents the time for PIToD, and the dashed line represents the estimated time for LOO. The left figure shows the time for both PIToD and LOO. The right figure shows the time for PIToD alone to allow readers to see the details of PIToD's time more clearly. The results show that the time required for LOO increases quadratically with the number of epochs, whereas the time required for PIToD increases linearly.

## 5.2 How efficiently does PIToD estimate the influence of experiences? Evaluation for computational time

We evaluate the computational time required for influence estimation with PIToD and compare it to the estimated time for LOO. To measure the computational time for PIToD, we run the method under the same settings as in the previous section and record its wall-clock time. For comparison, we also evaluate the estimated time required for influence estimation using LOO (Section 3). To estimate the time for LOO, we record the average time required for one policy iteration with PIToD and multiply this by the total number of policy iterations required for LOO [1].

The evaluation results (Figure 5) show that PIToD significantly reduces computational time compared to LOO. The time required for LOO increases quadratically as epochs progress, taking, for example, more than $4 \cdot 10^7$ seconds ($\approx 462$ days) up to 300 epochs in Humanoid. In contrast, the time required for PIToD increases linearly, taking about $1.4 \cdot 10^5$ seconds ($\approx$ one day) for 300 epochs in Humanoid.

## 6 Application of PIToD: amending policies and Q-functions by deleting negatively influential experiences

In the previous section, we demonstrated that PIToD can accurately and efficiently estimate the influence of experiences. What scenarios might benefit from this capability? In this section, we demonstrate how PIToD can be used to amend underperforming policies and Q-functions.

We amend policies and Q-functions by deleting experiences that negatively influence performance. We evaluate the performance of policies and Q-functions based respectively on returns and Q-estimation biases (Fujimoto et al., 2018; Chen et al., 2021a). The influence of an experience $e_i$ on the return, $L_{\text{ret}}$, is evaluated as follows:

$$L_{\text{ret}}(\pi_{\theta, \mathbf{w}_i}) - L_{\text{ret}}(\pi_\theta), \text{ where } L_{\text{ret}}(\pi) = \mathbb{E}_{a_t \sim \pi(\cdot|s_t)} \left[ \sum_{t=0}^{\infty} \gamma^t r(s_t, a_t) \right]. \tag{10}$$

Here, $s_t$ is sampled from an environment. In our setup, $L_{\text{ret}}$ is estimated using Monte Carlo returns collected by rolling out policies $\pi_{\theta, \mathbf{w}_i}$ and $\pi_\theta$. The influence of $e_i$ on Q-estimation bias, $L_{\text{bias}}$, is

---

[1] The total number of policy iterations for LOO is $I^2$, as discussed in Section 3. However, in the practical implementation of PIToD used in our experiments, we divide the experiences in the buffer into groups of 5000 experiences and estimate the influence of each group (Appendix C). For a fair comparison with this implementation, we use $\frac{I^2}{5000}$ instead of $I^2$ as the total number of policy iterations for LOO.

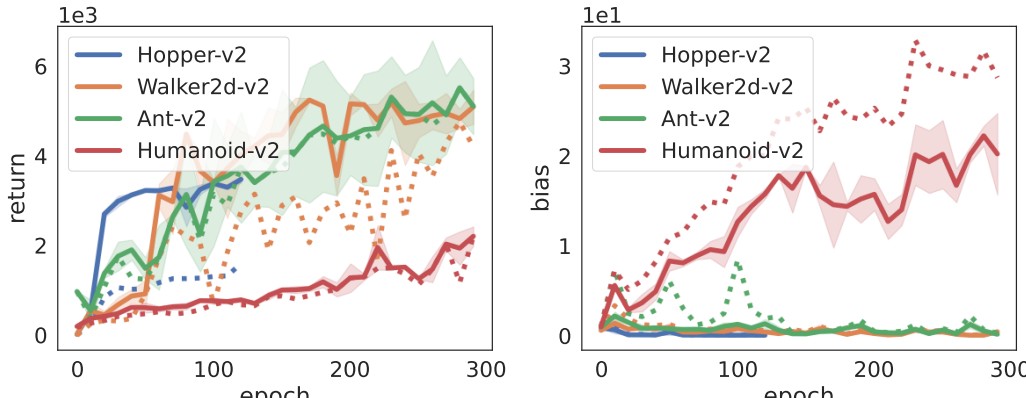

Figure 6: Results of policy amendments (left) and Q-function amendments (right) in underperforming trials. The solid lines represent the post-amendment performances: return for the policy (left; i.e., $L_{\text{ret}}(\pi_{\theta, \mathbf{w}_*})$) and bias for the Q-function (right; i.e., $L_{\text{bias}}(Q_{\phi, \mathbf{w}_*})$). The dashed lines show the pre-amendment performances: return (left; i.e., $L_{\text{ret}}(\pi_\theta)$) and bias (right; i.e., $L_{\text{bias}}(Q_\phi)$). These figures demonstrate that the amendments improve returns in Hopper and Walker2d, and reduce biases in Ant and Humanoid.

evaluated as follows:

$$L_{\text{bias}}(Q_{\phi, \mathbf{w}_i}) - L_{\text{bias}}(Q_\phi),$$

$$\text{where } L_{\text{bias}}(Q) = \mathbb{E}_{a_t \sim \pi_\theta(\cdot | s_t), a_{t'} \sim \pi_\theta(\cdot | s_{t'})} \left[ \sum_{t=0} \frac{\left| Q(s_t, a_t) - \sum_{t'=t} \gamma^{t'} r(s_{t'}, a_{t'}) \right|}{|\sum_{t'=t} \gamma^{t'} r(s_{t'}, a_{t'})|} \right]. \tag{11}$$

Here, $L_{\text{bias}}$ quantifies the discrepancy between the estimated and true Q-values using their L1 distance. Based on Eq. 10 and Eq. 11, we identify and delete the experience $e_*$ that has the strongest negative influence on them. We apply $\mathbf{w}_*$, which maximizes Eq. 10, to the policy to delete $e_*$. Additionally, we apply $\mathbf{w}_*$, which minimizes Eq. 11, to the Q-function to delete $e_*$. The algorithmic description of our amendment process is presented in Algorithm 4 in Appendix D.

We evaluate the effect of the amendments on trials in which the policy and Q-function underperform. We run ten learning trials with the amendments (Algorithm 4) and evaluate (i) $L_{\text{ret}}(\pi_{\theta, \mathbf{w}_*})$ for the two trials in which the policy scores the lowest returns $L_{\text{ret}}(\pi_\theta)$ and (ii) $L_{\text{bias}}(Q_{\phi, \mathbf{w}_*})$ for the two trials in which the Q-function scores the highest biases $L_{\text{bias}}(Q_\phi)$. The average scores of $L_{\text{ret}}(\pi_{\theta, \mathbf{w}_*})$ and $L_{\text{bias}}(Q_{\phi, \mathbf{w}_*})$ for these underperforming trials are shown in Figure 6. The average scores of $L_{\text{ret}}(\pi_{\theta, \mathbf{w}_*})$ and $L_{\text{bias}}(Q_{\phi, \mathbf{w}_*})$ for all ten trials are shown in Figure 11 in Appendix E.

The results of the policy and Q-function amendments (Figure 6) show that performance is improved through the amendments. From the policy amendment results (left part of Figure 6), we see that the return ($L_{\text{ret}}$) is significantly improved in Hopper and Walker. For example, in Hopper, the return before the amendment (the blue dashed line) is approximately 1000, but after the amendment (the blue solid line), it exceeds 3000. Additionally, from the Q-function amendment results (right part of Figure 6), we see that the Q-estimation bias ($L_{\text{bias}}$) is significantly reduced in Ant and Humanoid. For example, in Humanoid, the estimation bias of the Q-function before the amendment (the red dashed line) is approximately 30 during epochs 250–300, but after the amendment, it is reduced to approximately 20 (the red solid line).

What kinds of experiences negatively influence policy or Q-function performance? **Policy performance:** Some experiences negatively influencing returns are associated with stumbling or falling. An example of such experiences in Hopper is shown in the video "PIToD-Hopper.mp4," which is included in the supplementary material. **Q-function performance:** Experiences negatively influencing Q-estimation bias tend to be older experiences. The lower part of Figure 12 in Appendix E shows the distribution of influences on Q-estimation bias in each environment. For example, in the

Humanoid environment, we observe that older experiences often have a negative influence (highlighted in darker colors).

**Additional experiments.** We analyzed the correlation between the experience influences (i.e., Eq. 10 and Eq. 11) (Appendix F). Additionally, we performed amendments for other environments and RL agents using PIToD (Appendix G and Appendix H).

## 7 RELATED WORK

**Influence estimation in supervised learning.** Our research builds upon prior studies that estimate the influence of data within the supervised learning (SL) regime. In Section 4, we introduced our method for estimating the influence of data (i.e., experiences) in RL settings. Methods that estimate the influence of data have been extensively studied in the SL research community. Typically, these methods require SL loss functions that are twice differentiable with respect to model parameters (e.g., Koh & Liang (2017); Yeh et al. (2018); Hara et al. (2019); Koh et al. (2019); Guo et al. (2020); Chen et al. (2021b); Schioppa et al. (2022)). However, these methods are not directly applicable to our RL setting, as such SL loss functions are unavailable. In contrast, turn-over dropout (ToD) (Kobayashi et al., 2020) estimates the influence without requiring differentiable SL loss functions. We extended ToD for RL settings (Sections 4, 5, and 6). For this extension of ToD, we provided a theoretical justification (Appendix A) and considered practical implementations (Appendix C).

**Influence estimation in off-policy evaluation (OPE).** A few studies in the OPE community have focused on efficiently estimating the influence of experiences (Gottesman et al., 2020; Lobo et al., 2022). These studies are limited to estimating the influence on policy evaluation using nearest-neighbor or linear Q-functions. In contrast, our study estimates influence on a broader range of performance metrics (e.g., return or Q-estimation bias) using neural-network-based Q-functions and policies.

**Prioritized experience replay (PER).** In PER, the importance of experiences is estimated to prioritize experiences during experience replay. The importance of experiences is estimated based on criteria such as TD-error (Schaul et al., 2016; Fedus et al., 2020) or on-policyness (Novati & Koumoutsakos, 2019; Sun et al., 2020). Some readers might think that PER resembles our method. However, PER fundamentally differs from our method, as it cannot efficiently estimate or disable the influence of experiences in hindsight.

**Interpretable RL.** Our method (Section 4) estimates the influence of experiences, thereby providing a certain type of interpretability. Previous studies in the RL community have proposed interpretable methods based on symbolic (or relational) representation (Džeroski et al., 2001; Yang et al., 2018; Lyu et al., 2019; Garnelo et al., 2016; Andersen & Konidaris, 2017; Konidaris et al., 2018), interpretable proxy models (e.g., decision trees) (Degris et al., 2006; Liu et al., 2019; Coppens et al., 2019; Zhu et al., 2022), saliency explanation (Zahavy et al., 2016; Greydanus et al., 2018; Mott et al., 2019; Wang et al., 2020; Anderson et al., 2020), and sparse kernel models (Dao et al., 2018) [2]. Unlike these studies, our study proposes a method to estimate the influence of experiences on RL agent performance. This method helps us, for example, identify influential experiences when RL agents perform poorly, as demonstrated in Section 6.

## 8 CONCLUSION AND LIMITATIONS

In this paper, we proposed PIToD, a policy iteration (PI) method that efficiently estimates the influence of experiences (Section 4). We demonstrated that PIToD (i) accurately estimates the influence of experiences (Section 5.1), and (ii) significantly reduces the time required for influence estimation compared to the leave-one-out (LOO) method (Section 5.2). Furthermore, we applied PIToD to identify and delete negatively influential experiences, which improved the performance of policies and Q-functions (Section 6).

---

[2]For a comprehensive review of interpretable RL, see Milani et al. (2024).

We believe that our work provides a solid foundation for understanding the relationship between experiences and RL agent performance. However, it has several limitations. Details on these limitations and directions for future work are summarized in Appendix I.

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

## A  IMPORTANT THEORETICAL PROPERTY OF PIToD

In this section, we theoretically prove the following property of PIToD: "Assuming that the policy $\pi_\theta$ and the Q-function $Q_\phi$ are updated according to Algorithm 2, the functions $Q_{\phi,\mathbf{w}_i}$ and $\pi_{\theta,\mathbf{w}_i}$, which use the flipped mask $\mathbf{w}_i$, are unaffected by the gradients associated with experience $e_i$." This property is important as it justifies the use of the flipped mask $\mathbf{w}_i$ to estimate the influence of $e_i$ in PIToD.

First, we define key terms for our theoretical proof:
**Experience**: We define an experience $e_i$ as $e_i = (s, a, r, s', i)$, where $s$ is the state, $a$ is the action, $r$ is the reward, $s'$ is the next state, and $i$ is a unique identifier. We also define another experience as $e_{i'}$, where $i'$ is a unique identifier.
**Parameters**: At the $j$-th iteration of Algorithm 2 (lines 3–6), we define the parameters of the Q-function and policy that are not dropped by the mask $\mathbf{m}_{i'}$ as $\phi_{j,\mathbf{m}_{i'}}$ and $\theta_{j,\mathbf{m}_{i'}}$, respectively. Additionally, We define parameters that are dropped by $\mathbf{m}_{i'}$ as $\phi_{j,\mathbf{w}_{i'}}$ and $\theta_{j,\mathbf{w}_{i'}}$.
**Policy and Q-function**: We define the policy and Q-function, where all parameters except $\phi_{j,\mathbf{m}_{i'}}$ and $\theta_{j,\mathbf{m}_{i'}}$ are set to zero (i.e., dropped), as $Q_{\phi_{j,\mathbf{m}_{i'}}}$ and $\pi_{\theta_{j,\mathbf{m}_{i'}}}$. Similarly, the policy and Q-function, where all parameters except $\phi_{j,\mathbf{w}_{i'}}$ and $\theta_{j,\mathbf{w}_{i'}}$ are zero, are defined as $Q_{\phi_{j,\mathbf{w}_{i'}}}$ and $\pi_{\theta_{j,\mathbf{w}_{i'}}}$.

Next, we introduce two assumptions required for our proof. The first assumption is for the policy and Q-function with masks.

**Assumption 1.** $Q_{\phi_{j,\mathbf{m}_{i'}}}$ and $\pi_{\theta_{j,\mathbf{m}_{i'}}}$ can be replaced by $Q_{\phi'_{j,\mathbf{m}_{i'}}}$ and $\pi_{\theta'_{j,\mathbf{m}_{i'}}}$, whose parameters $\phi'_{j,\mathbf{m}_{i'}}$ and $\theta'_{j,\mathbf{m}_{i'}}$ satisfy the following gradient properties:

The property of $\phi'_{j,\mathbf{m}_{i'}}$ is as follows:

$$\nabla_{\phi'_{j,\mathbf{m}_{i'}}} \left( r + \gamma Q_{\bar{\phi}'_{j,\mathbf{m}_i}}(s', a') - Q_{\phi'_{j,\mathbf{m}_i}}(s, a) \right)^2, \quad a' \sim \pi_{\theta'_{j,\mathbf{m}_i}}(\cdot|s')$$
$$= \nabla_{\phi'_{j,\mathbf{m}_{i'}}} \left( r + \gamma Q_{\bar{\phi}'_{j,\mathbf{m}_i}}(s', a') - Q_{\phi'_{j,\mathbf{m}_i}}(s, a) \right)^2 \cdot \mathbb{I}(i = i'), \quad a' \sim \pi_{\theta'_{j,\mathbf{m}_i}}(\cdot|s').$$

Here, $\mathbb{I}$ is an indicator function that returns 1 if the specified condition (i.e., $i = i'$) is true and 0 otherwise.

The property of $\theta'_{j,\mathbf{m}_{i'}}$ is as follows:

$$\nabla_{\theta'_{j,\mathbf{m}_{i'}}} Q_{\phi'_{j+1,\mathbf{m}_i}}(s, a), \quad a \sim \pi_{\theta'_{j,\mathbf{m}_i}}(\cdot|s)$$
$$= \nabla_{\theta'_{j,\mathbf{m}_{i'}}} Q_{\phi'_{j+1,\mathbf{m}_i}}(s, a) \cdot \mathbb{I}(i = i'), \quad a \sim \pi_{\theta'_{j,\mathbf{m}_i}}(\cdot|s).$$

Intuitively, Assumption 1 can be interpreted as "$Q_{\phi_{j,\mathbf{m}_{i'}}}$ and $\pi_{\theta_{j,\mathbf{m}_{i'}}}$ are dominantly influenced by the experience $e_{i'}$ (i.e., the influence of other experiences is negligible)."

The second assumption is for $\phi_{j,\mathbf{w}_{i'}}$ and $\theta_{j,\mathbf{w}_{i'}}$:

**Assumption 2.** *For the gradient with respect to $\phi_{j,\mathbf{w}_{i'}}$, the following equation holds:*

$$\nabla_{\phi_{j,\mathbf{w}_{i'}}} \left( r + \gamma Q_{\bar{\phi}_{j,\mathbf{m}_i}}(s', a') - Q_{\phi_{j,\mathbf{m}_i}}(s, a) \right)^2, \quad a' \sim \pi_{\theta'_{j,\mathbf{m}_i}}(\cdot|s')$$
$$= \nabla_{\phi_{j,\mathbf{w}_{i'}}} \left( r + \gamma Q_{\bar{\phi}_{j,\mathbf{m}_i}}(s', a') - Q_{\phi_{j,\mathbf{m}_i}}(s, a) \right)^2 \cdot \mathbb{I}(i \neq i'), \quad a' \sim \pi_{\theta'_{j,\mathbf{m}_i}}(\cdot|s'). \quad (12)$$

*For the gradient with respect to $\theta_{j,\mathbf{w}_{i'}}$, the following equation holds:*

$$\nabla_{\theta_{j,\mathbf{w}_{i'}}} Q_{\phi_{j+1,\mathbf{m}_i}}(s, a), \quad a \sim \pi_{\theta_{j-1,\mathbf{m}_i}}(\cdot|s)$$
$$= \nabla_{\theta_{j,\mathbf{w}_{i'}}} Q_{\phi_{j+1,\mathbf{m}_i}}(s, a) \cdot \mathbb{I}(i \neq i'), \quad a \sim \pi_{\theta_{j,\mathbf{m}_i}}(\cdot|s). \quad (13)$$

Intuitively, Assumption 2 can be interpreted as "When updating parameters by using $e_i$, the parameters dropped out (i.e., $\phi_{j,\mathbf{w}_i}$ and $\theta_{j,\mathbf{w}_i}$) are not influenced by the gradient that is calculated with $e_i$."

Based on the above assumptions, we will derive the property of PIToD described at the beginning of this section [3]. Some readers may think that Assumption 2 corresponds to this property. However, in addition to Assumption 2, we must guarantee that the components used to create target signals for Eq. 12 and Eq. 13 (i.e., the components highlighted in red below) are also not influenced by $e_i$ when $i \neq i'$. Otherwise, $\phi_{j,\mathbf{w}_i}$ and $\theta_{j,\mathbf{w}_i}$ might still be updated by using components influenced by $e_i$ even when $i \neq i'$.

$$\nabla_{\phi_{j,\mathbf{w}_{i'}}} \left( r + \gamma Q_{\bar{\phi}_{j,\mathbf{m}_i}}(s',a') - Q_{\phi_{j,\mathbf{m}_i}}(s,a) \right)^2 \cdot \mathbb{I}(i \neq i'), \;\; a' \sim \pi_{\theta'_{j,\mathbf{m}_i}}(\cdot|s').$$

$$\nabla_{\theta_{j,\mathbf{w}_{i'}}} Q_{\phi_{j+1,\mathbf{m}_i}}(s,a) \cdot \mathbb{I}(i \neq i'), \;\; a \sim \pi_{\theta_{j,\mathbf{m}_i}}(\cdot|s).$$

Based on Assumption 1, we can ensure that these red-highlighted components are not influenced by $e_i$ when $i \neq i'$.

Based on Assumption 1, the following theorem holds:

**Theorem 1.** *Given that, for $j > 0$, the parameters $\phi'_{j,\mathbf{m}_{i'}}$ and $\theta'_{j,\mathbf{m}_{i'}}$ are updated in the same way as the original parameters $\phi_{j,\mathbf{m}_{i'}}$ and $\theta_{j,\mathbf{m}_{i'}}$, according to Eq. 5 and Eq. 6, the following equation holds:*

$$\phi'_{j,\mathbf{m}_{i'}} \;\; \leftarrow \;\; \phi'_{j-1,\mathbf{m}_{i'}} - \sum_{(s,a,r,s',i)} \nabla_{\phi'_{j-1,\mathbf{m}_{i'}}} \left( r + \gamma Q_{\bar{\phi}'_{j-1,\mathbf{m}_i}}(s',a') - Q_{\phi'_{j-1,\mathbf{m}_i}}(s,a) \right)^2 \cdot \mathbb{I}(i = i'),$$

$$a' \sim \pi_{\theta'_{j-1,\mathbf{m}_i}}(\cdot|s').$$

$$\theta'_{j,\mathbf{m}_{i'}} \leftarrow \theta'_{j-1,\mathbf{m}_{i'}} - \sum_{(s,a,r,s',i)} \nabla_{\theta'_{j-1,\mathbf{m}_{i'}}} Q_{\phi'_{j,\mathbf{m}_i}}(s,a) \cdot \mathbb{I}(i = i'), \;\; a \sim \pi_{\theta'_{j-1,\mathbf{m}_i}}(\cdot|s).$$

*Proof.*

$$\phi'_{j,\mathbf{m}_{i'}} \;\; \leftarrow \;\; \phi'_{j-1,\mathbf{m}_{i'}} - \nabla_{\phi'_{j-1,\mathbf{m}_{i'}}} \sum_{(s,a,r,s',i)} \left( r + \gamma Q_{\bar{\phi}'_{j-1,\mathbf{m}_i}}(s',a') - Q_{\phi'_{j-1,\mathbf{m}_i}}(s,a) \right)^2,$$

$$a' \sim \pi_{\theta'_{j-1,\mathbf{m}_i}}(\cdot|s')$$

$$\overset{(1)}{=} \;\; \phi'_{j-1,\mathbf{m}_{i'}} - \sum_{(s,a,r,s',i)} \nabla_{\phi'_{j-1,\mathbf{m}_{i'}}} \left( r + \gamma Q_{\bar{\phi}'_{j-1,\mathbf{m}_i}}(s',a') - Q_{\phi'_{j-1,\mathbf{m}_i}}(s,a) \right)^2 \cdot \mathbb{I}(i = i'),$$

$$a' \sim \pi_{\theta'_{j-1,\mathbf{m}_i}}(\cdot|s')$$

$$\theta'_{j,\mathbf{m}_{i'}} \;\; \leftarrow \;\; \theta'_{j-1,\mathbf{m}_{i'}} - \nabla_{\theta'_{j-1,\mathbf{m}_{i'}}} \sum_{(s,a,r,s',i)} Q_{\phi'_{j,\mathbf{m}_i}}(s,a), \;\; a \sim \pi_{\theta'_{j-1,\mathbf{m}_i}}(\cdot|s)$$

$$\overset{(1)}{=} \;\; \theta'_{j-1,\mathbf{m}_{i'}} - \sum_{(s,a,r,s',i)} \nabla_{\theta'_{j-1,\mathbf{m}_{i'}}} Q_{\phi'_{j,\mathbf{m}_i}}(s,a) \cdot \mathbb{I}(i = i'), \;\; a \sim \pi_{\theta'_{j-1,\mathbf{m}_i}}(\cdot|s)$$

(1) Apply Assumption 1. □

This theorem implies that $Q_{\phi'_{j,\mathbf{m}_{i'}}}$ and $\pi_{\theta'_{j,\mathbf{m}_{i'}}}$ are dominantly influenced by the experience $e_{i'}$ for $j > 0$. Thus, if the red-highlighted components above can be replaced with these components, we can say that $\phi_{j,\mathbf{w}_i}$ and $\theta_{j,\mathbf{w}_i}$ are not influenced by gradients depending on $e_i$ in both cases of $i = i'$ and $i \neq i'$. Below, we will show that such a replacement is doable.

Based on Assumptions 1 and 2, the following theorem holds:

---

[3]"Assuming that the policy $\pi_\theta$ and the Q-function $Q_\phi$ are updated according to Algorithm 2, the functions $Q_{\phi,\mathbf{w}_i}$ and $\pi_{\theta,\mathbf{w}_i}$, which use the flipped mask $\mathbf{w}_i$, are unaffected by the gradients associated with experience $e_i$."

**Theorem 2.** *For any $j > 0$, the parameters $\phi_{j,\mathbf{w}_{i'}}$ and $\theta_{j,\mathbf{w}_{i'}}$ in Algorithm 2 are updated as follows:*

$$\phi_{j,\mathbf{w}_{i'}} \quad \leftarrow \quad \phi_{j-1,\mathbf{w}_{i'}} - \sum_{(s,a,r,s',i)} \nabla_{\phi_{j-1,\mathbf{w}_{i'}}} \left( r + \gamma Q_{\bar{\phi}'_{j-1,\mathbf{m}_i}}(s',a') - Q_{\phi_{j-1,\mathbf{m}_i}}(s,a) \right)^2 \cdot \mathbb{I}(i \neq i'),$$
$$a' \sim \pi_{\theta'_{j-1,\mathbf{m}_i}}(\cdot|s')$$

$$\theta_{j,\mathbf{w}_{i'}} \quad \leftarrow \quad \theta_{j-1,\mathbf{w}_{i'}} - \sum_{(s,a,r,s',i)} \nabla_{\theta_{j-1,\mathbf{w}_{i'}}} Q_{\phi'_{j,\mathbf{m}_i}}(s,a) \cdot \mathbb{I}(i \neq i'), \quad a \sim \pi_{\theta_{j-1,\mathbf{m}_i}}(\cdot|s)$$

*Proof.* For $\phi_{j,\mathbf{w}_{i'}}$,

$$\phi_{j,\mathbf{w}_{i'}} \quad \leftarrow \quad \phi_{j-1,\mathbf{w}_{i'}} - \nabla_{\phi_{j-1,\mathbf{w}_{i'}}} \sum_{(s,a,r,s',i)} \left( r + \gamma Q_{\bar{\phi}_{j-1,\mathbf{m}_i}}(s',a') - Q_{\phi_{j-1,\mathbf{m}_i}}(s,a) \right)^2,$$
$$a' \sim \pi_{\theta_{j-1,\mathbf{m}_i}}(\cdot|s')$$

$$\overset{(1)}{=} \quad \phi_{j-1,\mathbf{w}_{i'}} - \sum_{(s,a,r,s',i)} \nabla_{\phi_{j-1,\mathbf{w}_{i'}}} \left( r + \gamma Q_{\bar{\phi}_{j-1,\mathbf{m}_i}}(s',a') - Q_{\phi_{j-1,\mathbf{m}_i}}(s,a) \right)^2 \cdot \mathbb{I}(i \neq i'),$$
$$a' \sim \pi_{\theta_{j-1,\mathbf{m}_i}}(\cdot|s')$$

$$\overset{(2)}{=} \quad \phi_{j-1,\mathbf{w}_{i'}} - \sum_{(s,a,r,s',i)} \nabla_{\phi_{j-1,\mathbf{w}_{i'}}} \left( r + \gamma Q_{\bar{\phi}'_{j-1,\mathbf{m}_i}}(s',a') - Q_{\phi_{j-1,\mathbf{m}_i}}(s,a) \right)^2 \cdot \mathbb{I}(i \neq i'),$$
$$a' \sim \pi_{\theta'_{j-1,\mathbf{m}_i}}(\cdot|s')$$

(1) Apply Assumption 2. (2) Apply Assumption 1.

Similarly, for $\theta_{j,\mathbf{w}_{i'}}$,

$$\theta_{j,\mathbf{w}_{i'}} \quad \leftarrow \quad \theta_{j-1,\mathbf{w}_{i'}} - \nabla_{\theta_{j-1,\mathbf{w}_{i'}}} \sum_{(s,a,r,s',i)} Q_{\phi_{j,\mathbf{m}_i}}(s,a), \quad a \sim \pi_{\theta_{j-1,\mathbf{m}_i}}(\cdot|s)$$

$$\overset{(1)}{=} \quad \theta_{j-1,\mathbf{w}_{i'}} - \sum_{(s,a,r,s',i)} \nabla_{\theta_{j-1,\mathbf{w}_{i'}}} Q_{\phi_{j,\mathbf{m}_i}}(s,a) \cdot \mathbb{I}(i \neq i'), \quad a \sim \pi_{\theta_{j-1,\mathbf{m}_i}}(\cdot|s)$$

$$\overset{(2)}{=} \quad \theta_{j-1,\mathbf{w}_{i'}} - \sum_{(s,a,r,s',i)} \nabla_{\theta_{j-1,\mathbf{w}_{i'}}} Q_{\phi'_{j,\mathbf{m}_i}}(s,a) \cdot \mathbb{I}(i \neq i'), \quad a \sim \pi_{\theta_{j-1,\mathbf{m}_i}}(\cdot|s)$$

$\square$

This theorem implies that:

(i) When $i = i'$, neither $\theta_{j,\mathbf{w}_{i'}}$ nor $\phi_{j,\mathbf{w}_{i'}}$ is influenced by gradients dependent on experience $e_{i'}$.

(ii) When $i \neq i'$, $\theta_{j,\mathbf{w}_{i'}}$ and $\phi_{j,\mathbf{w}_{i'}}$ are updated without depending on the components that might be influenced by $e_{i'}$.

Therefore, we conclude that "$Q_{\phi,\mathbf{w}_{i'}}$ and $\pi_{\theta,\mathbf{w}_{i'}}$, and consequently $Q_{\phi,\mathbf{w}_i}$ and $\pi_{\theta,\mathbf{w}_i}$, are not influenced by the gradients related to the experiences $e_{i'}$ and $e_i$, respectively."

# B   ANALYZING AND MINIMIZING OVERLAP IN ELEMENTS OF MASKS

In our method (Section 4), each experience is assigned a mask. If there is significant overlap in the elements of different masks, one experience could significantly interfere with other experiences. In this section, we discuss (i) the expected overlap between the masks of experiences $e_i$ and $e_{i'}$ and (ii) the dropout rate that minimizes this overlap.

For discussion, we introduce the following definitions and assumptions. We define the mask size as $M$, and the number of overlapping elements between masks as $m$. We assume that each mask

element is independently initialized as 0 with probability $p$ (i.e., dropout rate) and 1 with probability $1 - p$.

Below, we derive the probability and expected number of overlaps in the mask elements.

**Probability of** $m$ **overlaps.** First, we calculate the probability that a specific position in the masks of $e_i$ and $e_{i'}$ has the same value. The probability that both elements of the masks have 0 at the same position is $p \cdot p = p^2$. Similarly, the probability that both elements have 1 at the same position is $(1-p) \cdot (1-p) = (1-p)^2$. Therefore, the probability $q$ that the values at a specific position in the masks are the same is

$$q = p^2 + (1-p)^2 = 2p^2 - 2p + 1. \tag{14}$$

The probability that the masks have $m$ overlaps follows the binomial distribution:

$$\binom{M}{m} q^m (1-q)^{M-m}. \tag{15}$$

**Expected number of overlaps.** Using Eq.14 and Eq.15, the expected number of overlaps can be represented as

$$\sum_{k=0}^{M} k \binom{M}{k} q^k (1-q)^{M-k} = Mq \tag{16}$$
$$= M(2p^2 - 2p + 1).$$

For better understanding, we show a plot of Eq. 16 values with respect to $p$ and $M$ in Figure 7.

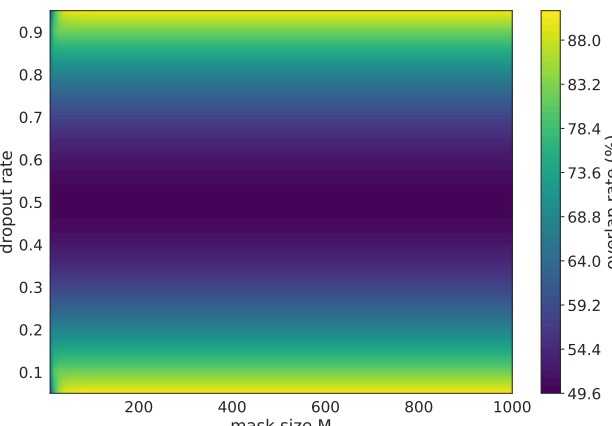

Figure 7: The distribution of the expected number of overlaps (Eq. 16) with respect to the dropout rate $p$ and mask size $M$. For clarity, we plot the expected overlap rate ($m/M$) instead of the expected number of overlaps $m$.

**The dropout rate of** $p = 0.5$ **minimizes the expected number of overlaps.** Since Eq. 16 is convex in $p$, the value of $p$ that minimizes the expected overlap is determined by solving $\frac{\mathrm{d}M(2p^2-2p+1)}{\mathrm{d}p} = 0$. As a result, we find that $p = 0.5$ minimizes the expected overlap. With $p = 0.5$, we can expect a 50% overlap between the two masks. Figure 8 shows the probability of the overlap rate $m/M$ with $p = 0.5$ for various values of $M$. From this figure, we see that the probability of having a between 0-50% overlap is very high, while the probability of having a between 50-100% overlap is very low, regardless of the value of $M$.

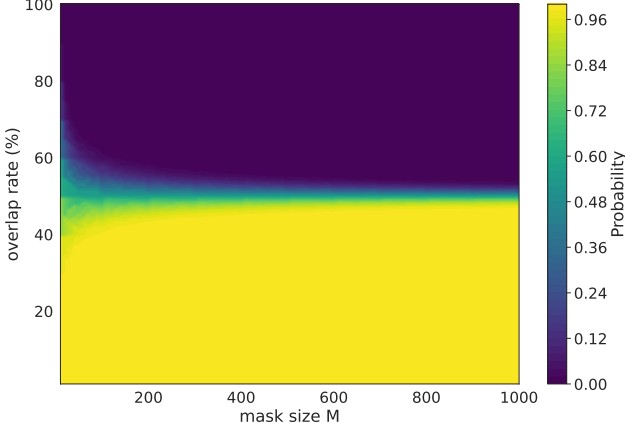

Figure 8: The probability of the overlap rate $m/M$ with $p = 0.5$ for various values of $M$.

---

**Algorithm 3** SAC version of PI with group mask in PIToD

---

1: Initialize policy parameters $\theta$, Q-function parameters $\phi_1$, $\phi_2$, and an empty replay buffer $\mathcal{B}$.
2: **for** $i' = 0, ..., I$ **do**
3:     Take action $a \sim \pi_\theta(\cdot|s)$; Observe reward $r$ and next state $s'$; Define an experience using the group identifier $i'' \leftarrow \lfloor i'/5000 \rfloor$ as $e_{i''} = (s, a, r, s', i'')$; $\mathcal{B} \leftarrow \mathcal{B} \bigcup \{e_{i''}\}$.
4:     Sample experiences $\{(s, a, r, s', i), ...\}$ from $\mathcal{B}$ (Here, $e_i = (s, a, r, s', i)$).
5:     Compute target $y_i$:
$$y_i = r + \gamma \left( \min_{j=1,2} Q_{\bar{\phi}_j, \mathbf{m}_i}(s', a') - \alpha \log \pi_{\theta, \mathbf{m}_i}(a'|s') \right), \quad a' \sim \pi_{\theta, \mathbf{m}_i}(\cdot|s').$$
6:     **for** $j = 1, 2$ **do**
7:         Update $\phi_j$ with gradient descent using
$$\nabla_{\phi_j} \sum_{(s,a,r,s',i)} \left( Q_{\phi_j, \mathbf{m}_i}(s, a) - y_i \right)^2.$$
8:         Update target networks with $\bar{\phi}_j \leftarrow \rho \bar{\phi}_j + (1 - \rho)\phi_j$.
9:     Update $\theta$ with gradient ascent using
$$\nabla_\theta \sum_{(s,a,r,s',i)} \left( \frac{1}{2} \sum_{i=1}^{2} Q_{\phi_j, \mathbf{m}_i}(s, a_{\theta, \mathbf{m}_i}) - \alpha \log \pi_{\theta, \mathbf{m}_i}(a|s) \right), \quad a, a_{\theta, \mathbf{m}_i} \sim \pi_{\theta, \mathbf{m}_i}(\cdot|s).$$

---

# C   PRACTICAL IMPLEMENTATION OF PIToD FOR SECTION 5 AND SECTION 6

In this section, we describe the practical implementation of PIToD. Specifically, we explain (i) the soft actor-critic (SAC) (Haarnoja et al., 2018b) version of PI with a mask, (ii) group mask, and (iii) key implementation decisions to improve learning. This practical implementation is used in our experiments (Section 5 and Section 6).

**(i) SAC version of PI with a mask.** The SAC version of PI with masks is presented in Algorithm 3. The mask is applied to the policy and Q-functions during policy evaluation (lines 5–8) and policy improvement (line 9). For the policy evaluation, two Q-functions $Q_{\phi_j}$, where $j \in \{1, 2\}$, are updated as:

$$
\begin{aligned}
\phi_j \leftarrow \phi_j \\
- \nabla_{\phi_j} \mathbb{E}_{e_i=(s,a,r,s',i)\sim\mathcal{B}, \, a'\sim\pi_{\theta,\mathbf{m}_i}(\cdot|s')} \Bigg[ \bigg( r + \gamma \left( \min_{j'=1,2} Q_{\bar{\phi}_{j'}, \mathbf{m}_i}(s', a') - \alpha \log \pi_{\theta, \mathbf{m}_i}(a'|s') \right) \\
- Q_{\phi_j, \mathbf{m}_i}(s, a) \bigg)^2 \Bigg].
\end{aligned}
$$
(17)

This is a variant of Eq. 1 that uses clipped double Q-learning with two target Q-functions $Q_{\bar{\phi}_{j'}, \mathbf{m}_i}$ and entropy bonus $\alpha \log \pi_{\theta, \mathbf{m}_i}(a'|s')$. Additionally, for policy improvement, policy $\pi_\theta$ is updated as

$$
\theta \leftarrow \theta + \nabla_\theta \mathbb{E}_{e_i=(s,i)\sim\mathcal{B}, \, a_{\theta,\mathbf{m}_i},a\sim\pi_{\theta,\mathbf{m}_i}(\cdot|s)} \left[ \left( \frac{1}{2} \sum_{j=1}^{2} Q_{\phi_j, \mathbf{m}_i}(s, a_{\theta, \mathbf{m}_i}) - \alpha \log \pi_{\theta, \mathbf{m}_i}(a|s) \right) \right].
$$
(18)

This is a variant of Eq. 2 that uses the entropy bonus.

**(ii) Group Mask.** In our preliminary experiments, we found that the influence of a single experience on performance was negligibly small. To examine more significant influences, we shifted our focus from the influence of individual experiences to grouped experiences. To estimate the influence of grouped experiences, we organize experiences into groups and assign a mask to each group. Specifically, we treated 5000 experiences as a single group. This grouping process was implemented by assigning a group identifier to each experience, calculated as $i'' \leftarrow \lfloor i'/5000 \rfloor$ (line 3 of Algorithm 3).

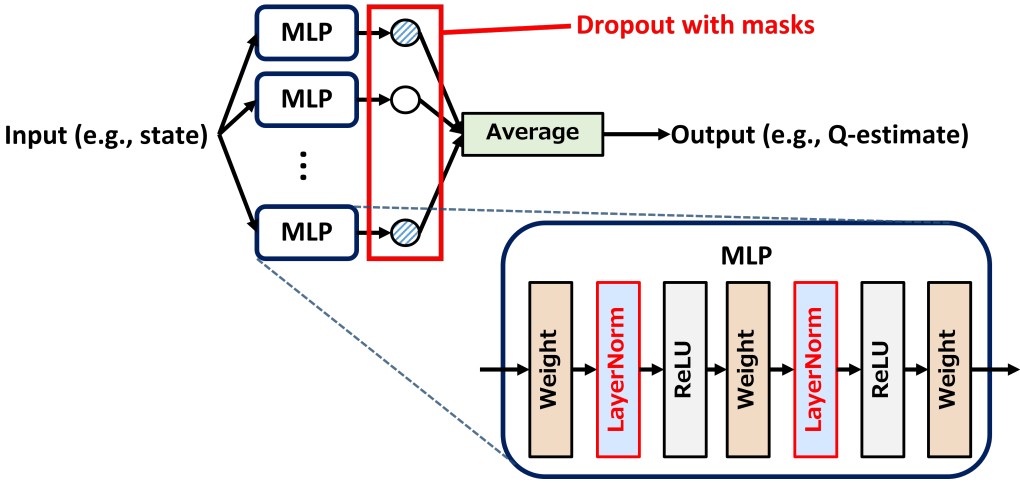

Figure 9: Network architectures for policy and Q-function. The policy network takes states as inputs and outputs the parameters of the policy distribution (mean and variance for a Gaussian distribution). The Q-function network takes state-action pairs as inputs and outputs Q-estimates. These networks incorporate macro-block dropout and layer normalization. **Macro-block dropout.** Our architecture utilizes an ensemble of 20 multi-layer perceptrons (MLPs), applying dropout with masks (and flipped masks) to each MLP's output. **Layer normalization.** Layer normalization is applied after every activation (ReLU) layer in each MLP.

**(iii) Key implementation decisions to improve learning.** In our preliminary experiments, we found that directly applying masks and flipped masks to dropping out the parameters of the policy and Q-function degrades learning performance. To address this issue, we implemented macro-block dropout and layer normalization (Figure 9). **Macro-block dropout.** Instead of applying dropout to individual parameters, we apply dropout at the block level. Specifically, we group several parameters into a "block" and apply dropout to these blocks. In our experiment, we used an ensemble of 20 multi-layer perceptrons (MLPs) for the policy and Q-function, and treated each MLP's parameters as a single block. **Layer normalization.** We applied layer normalization (Ba et al., 2016) after each activation (ReLU) layer. Recent works show that layer normalization improves learning in a wide range of RL settings (e.g., Hiraoka et al. (2022); Ball et al. (2023); Nauman et al. (2024)).

To evaluate the effect of our key implementation decisions, we compare four implementations of Algorithm 3:

**1. PIToD** applies vanilla dropout with masks to each parameter of the policy and Q-function.

**2. PIToD+LN** applies layer normalization to the policy and Q-function.

**3. PIToD+MD** applies macro-block dropout to the policy and Q-function.

**4. PIToD+LN+MD** applies layer normalization and macro-block dropout to the policy and Q-function.

These implementations are compared based on the empirical returns obtained in test episodes.

The comparison results (Figure 10) indicate that the implementation with our key decisions (PIToD+LN+MD) achieves the highest returns in each environment.

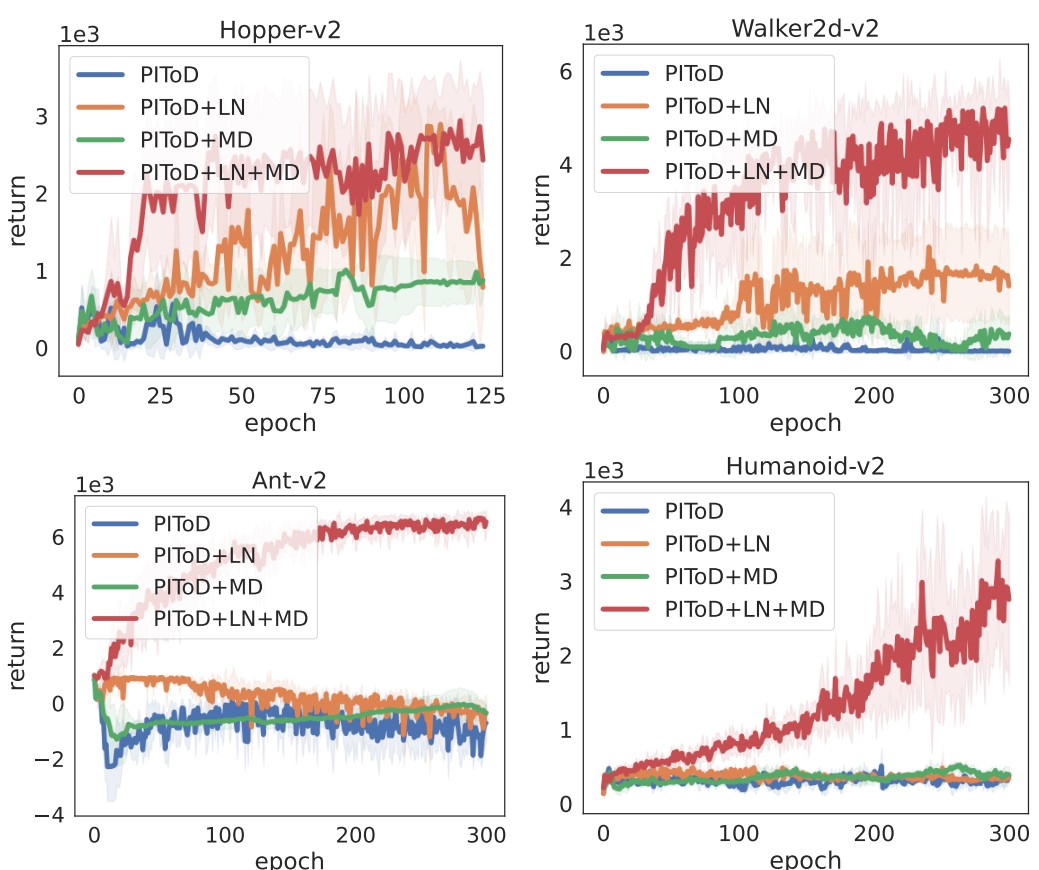

Figure 10: Ablation study results. The vertical axis represents returns, and the horizontal axis represents epochs. In each environment, the implementation with our key decisions (PIToD+LN+MD) achieves the highest returns.

# D    Algorithm for amending policy and Q-function used in Section 6

---

**Algorithm 4** Amendment of policy and Q-function using influence estimates. Lines 5–7 are for policy amendment. Lines 8–10 are for Q-function amendment.

---

1: Initialize policy parameters $\theta$, Q-function parameters $\phi$, and an empty replay buffer $\mathcal{B}$. Set the influence estimation interval $I_{\text{ie}}$.

2: **for** $i' = 0, ..., I$ iterations **do**

3:    Execute environment interaction, store experiences, and perform policy iteration as per lines 3–6 of Algorithm 2.

4:    **if** $i'\%I_{\text{ie}} = 0$ **then**

5:        Identify $\mathbf{w}_*$ for policy as follows:

$$\mathbf{w}_* = \arg\max_{\mathbf{w}_i} L_{\text{ret}}\left(\pi_{\theta,\mathbf{w}_i}\right) - L_{\text{ret}}\left(\pi_\theta\right).$$

6:        **if** $L_{\text{ret}}\left(\pi_{\theta,\mathbf{w}_*}\right) - L_{\text{ret}}\left(\pi_\theta\right) > 0$ **then**

7:           Evaluate the return of the amended policy $L_{\text{ret}}\left(\pi_{\theta,\mathbf{w}_*}\right)$.

8:        Identify $\mathbf{w}_*$ for Q-function as follows:

$$\mathbf{w}_* = \arg\min_{\mathbf{w}_i} L_{\text{bias}}\left(Q_{\phi,\mathbf{w}_i}\right) - L_{\text{bias}}\left(Q_\phi\right).$$

9:        **if** $L_{\text{bias}}\left(Q_{\phi,\mathbf{w}_*}\right) - L_{\text{bias}}\left(Q_\phi\right) < 0$ **then**

10:       Evaluate the Q-estimation bias of the amended Q-function $L_{\text{bias}}\left(Q_{\phi,\mathbf{w}_*}\right)$.

---

# E  SUPPLEMENTARY EXPERIMENTAL RESULTS FOR SECTION 6

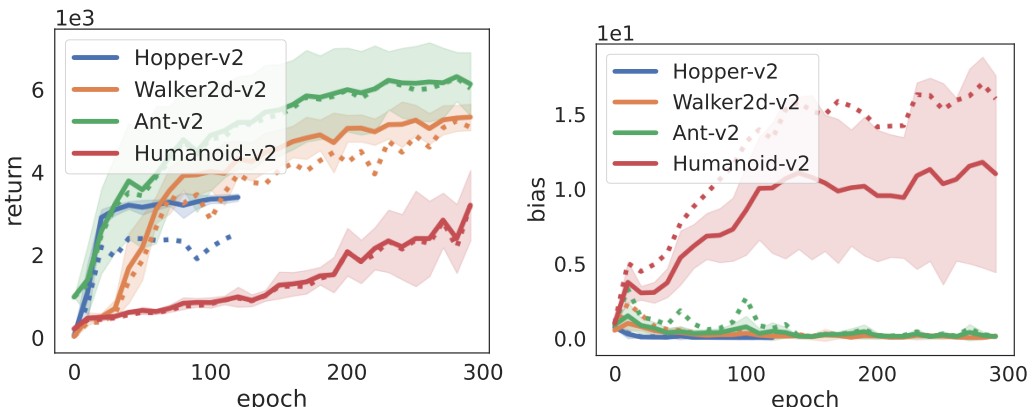

Figure 11: Results of policy amendments (left) and Q-function amendments (right) for all ten trials. The solid lines represent the post-amendment performances: return for the policy (left; i.e., $L_{\mathrm{ret}}(\pi_{\theta,\mathbf{w}_*})$) and bias for the Q-function (right; i.e., $L_{\mathrm{bias}}(Q_{\phi,\mathbf{w}_*})$). The dashed lines show the pre-amendment performances: return (left; i.e., $L_{\mathrm{ret}}(\pi_\theta)$) and bias (right; i.e., $L_{\mathrm{bias}}(Q_\phi)$).

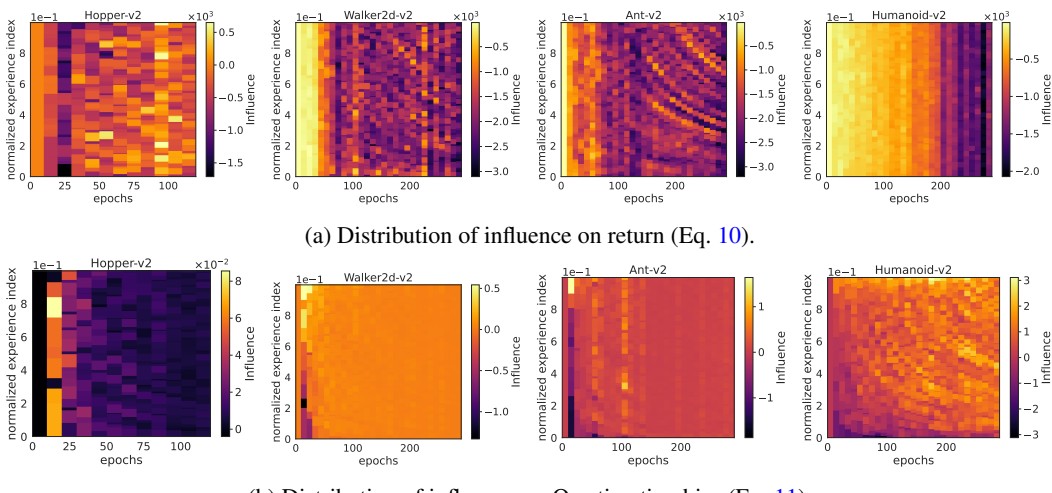

(a) Distribution of influence on return (Eq. 10).

(b) Distribution of influence on Q-estimation bias (Eq. 11).

Figure 12: Distribution of influence on return and Q-estimation bias for all ten trials. The vertical axis represents the normalized experience index, which ranges from 0.0 for the oldest experiences to 1.0 for the most recent experiences. The horizontal axis represents the number of epochs. The color bar represents the value of influence.

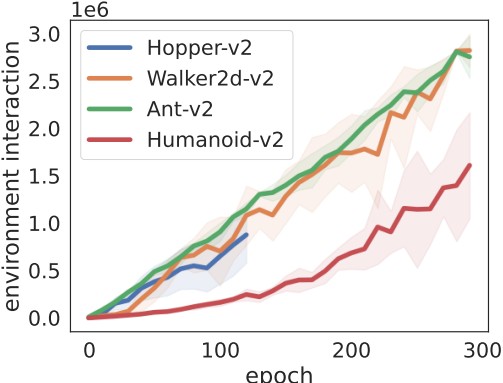

Figure 13: The number of environment interactions required for policy amendments in Section 6.

# F  ANALYSIS OF THE CORRELATION BETWEEN THE INFLUENCES OF EXPERIENCES

In Sections 5 and 6, we estimated the influences of experiences on performance (e.g., return or Q-estimation bias). In Appendix B, we discussed how the dropout rate of masks elements relates to the overlap between the masks. In this section, we analyze two points: (i) the correlation between the influences of experiences within each performance metric, and (ii) how the dropout rate of masks affects this correlation [4].

We calculate the correlation between the experience influences for each performance metric used in Sections 5 and 6. In these sections, we estimated the influences of experiences on policy evaluation ($L_{pe,i}$), policy improvement ($L_{pi,i}$), return ($L_{ret}$), and Q-estimation bias ($L_{bias}$). We treat the influences of experiences on each metric at each epoch as a vector of random variables, where each element represents the influence of a single experience. We calculate the Pearson correlation between these elements. The influence values observed in the ten learning trials are used as samples. In the following discussion, we focus on the average value of the correlations between the pairs of vector elements.

**(i) The correlation between the influences of experiences.** The correlation between the influences of experiences is shown in Figure 14. The figure shows that the correlation tends to approach zero as the number of epochs increases. For return and bias, the correlation converges to zero early in the learning process, regardless of the environments. For policy evaluation and improvement, the degree of correlation convergence varies significantly across environments.

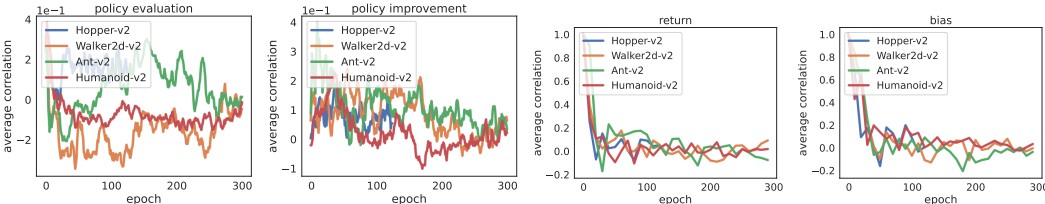

Figure 14: Correlation between the influences of experiences on policy evaluation ($L_{pe,i}$), policy improvement ($L_{pi,i}$), return ($L_{ret}$), and Q-estimation bias ($L_{bias}$) for each epoch in each environment. The vertical axis represents the average correlation of experience influences, ranging from -1.0 to 1.0. The horizontal axis represents the number of epochs.

**(ii) The relationship between the correlation and the dropout rate.** We evaluated the correlations between the influences of experiences by varying the dropout rate of the masks. Specifically, we evaluated the correlations using PIToD with four different dropout rates:
**DR0.5:** PIToD with a dropout rate of 0.5, which is the setting used in the main experiments of this paper.
**DR0.25:** PIToD with a dropout rate of 0.25.
**DR0.1:** PIToD with a dropout rate of 0.1.
**DR0.05:** PIToD with a dropout rate of 0.05.
The correlations for these cases in the Hopper environment are shown in Figure 15. The results imply that the impact of the dropout rate on the correlation depends significantly on the specific performance metric. For instance, we do not observe a significant impact of the dropout rate in policy evaluation or policy improvement. In contrast, for return, we observe that the correlation increases as the dropout rate decreases.

---

[4]Note that we focus on analyzing the correlation independently for each performance metric and do not examine correlations across different metrics.

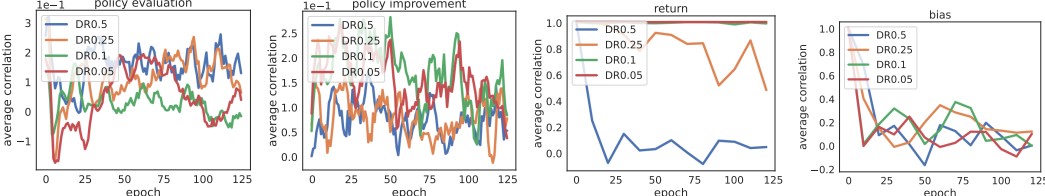

Figure 15: Correlation between the influences of experiences at each epoch in the Hopper environment. The vertical axis represents the average correlation of experience influences. The horizontal axis represents the number of learning epochs. Each label in the legend corresponds to a dropout rate for masks. For example, "DR0.5" means a dropout rate of $0.5$ (half of the elements in each mask are set to zero), and "DR0.1" means a dropout rate of $0.1$ ($10\%$ of the elements in each mask are set to zero).

# G  AMENDING POLICIES AND Q-FUNCTIONS IN DM CONTROL ENVIRONMENTS WITH ADVERSARIAL EXPERIENCES

In Section 6, we applied PIToD to amend policies and Q-functions in the MuJoCo (Todorov et al., 2012) environments.

In this section, we apply PIToD to amend policies and Q-functions in DM control (Tunyasuvunakool et al., 2020) environments with adversarial experiences. We focus on the DM control environments: finger-turn_hard, hopper-stand, hopper-hop, fish-swim, cheetah-run, quadruped-run, humanoid-run, and humanoid-stand. In these environments, we introduce adversarial experiences. An adversarial experience contains an adversarial reward $r'$, which is a reversed and magnified version of the original reward $r$: $r' = -100 \cdot r$. These adversarial experiences are designed to (i) disrupt the agent's ability to maximize original rewards and (ii) have greater influence than other non-adversarial experiences stored in the replay buffer. At 150 epochs (i.e., in the middle of training), the RL agent encounters 5000 adversarial experiences. In these environments, we amend policies and Q-functions as in Section 6.

The results of the policy and Q-function amendments (Figures 16 and 17) show that performance is improved by the amendments. The policy amendment results (Figure 16) show that returns are improved, particularly in fish-swim. Additionally, the Q-function amendment results (Figure 17) show that the Q-estimation bias is significantly reduced in finger-turn_hard, hopper-stand, hopper-hop, fish-swim, cheetah-run, and quadruped-run.

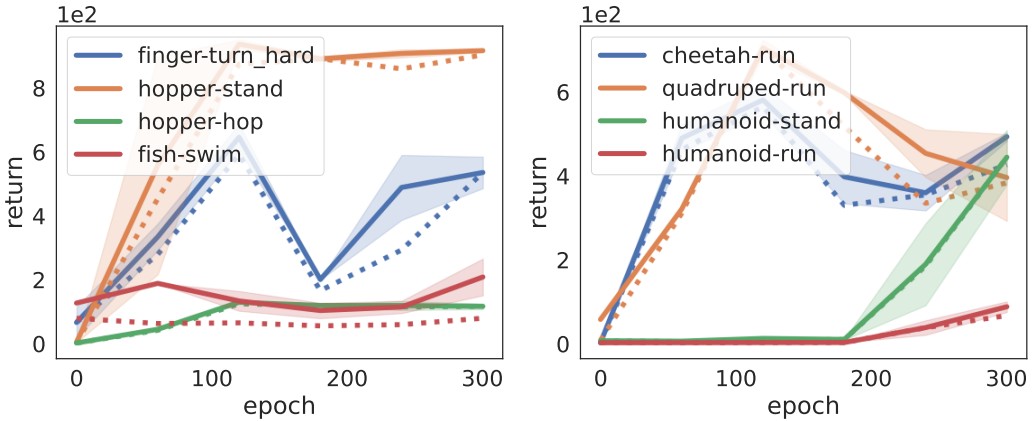

Figure 16: Results of policy amendments in DM control environments with adversarial experiences. The solid lines represent the post-amendment return for the policy (i.e., $L_{\text{ret}}(\pi_{\theta, \mathbf{w}_*})$). The dashed lines show the pre-amendment return (i.e., $L_{\text{ret}}(\pi_\theta)$).

**Can PIToD identify adversarial experiences?**  PIToD identifies adversarial experiences as (i) strongly influential experiences for policy evaluation and (ii) positively influential experiences for Q-estimation bias. **Policy evaluation:** Figure 18 shows the distribution of influences on policy evaluation. We observe that adversarial experiences have a strong influence (highlighted in lighter colors), except in humanoid-run. **Q-estimation bias:** Figure 19 shows the distribution of influences on Q-estimation bias. Interestingly, we observe that adversarial experiences have a strong positive influence (highlighted in lighter colors). Namely, these adversarial experiences contribute to reducing Q-estimation bias. However, after introducing adversarial experiences (i.e., after epoch 150), we also observe experiences with a negative influence. We hypothesize that adversarial experiences hinder the learning from other experiences.

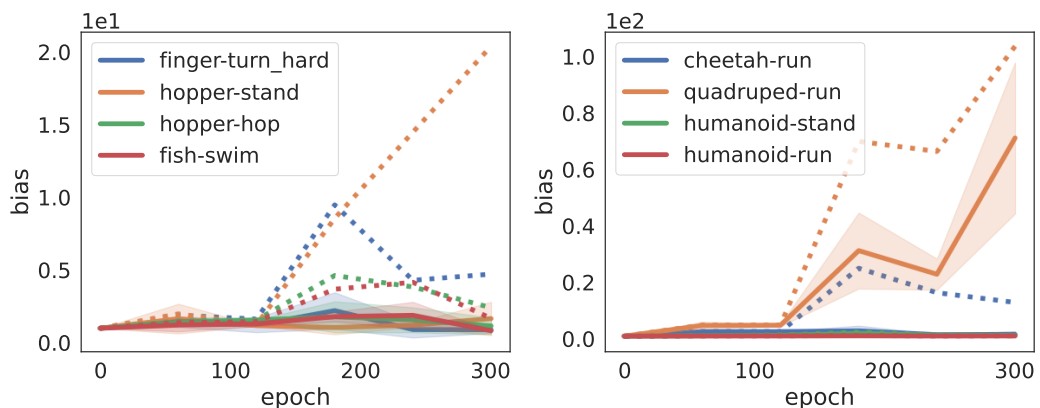

Figure 17: Results of Q-function amendments in DM control environments with adversarial experiences. The solid lines represent the post-amendment bias for the Q-function (i.e., $L_{\text{bias}}(Q_{\phi,\mathbf{w}_*})$). The dashed lines show the pre-amendment bias (i.e., $L_{\text{bias}}(Q_\phi)$).

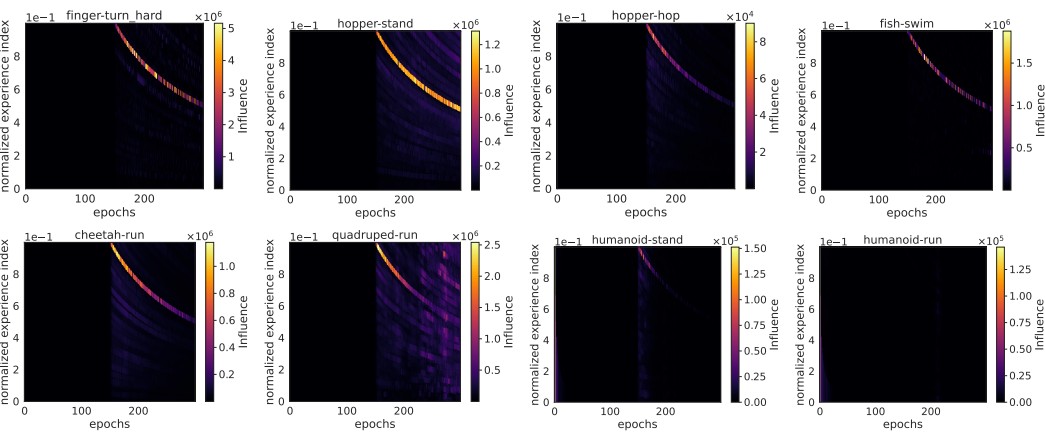

Figure 18: Distribution of influence on policy evaluation (Eq. 8) in DM control environments with adversarial experiences.

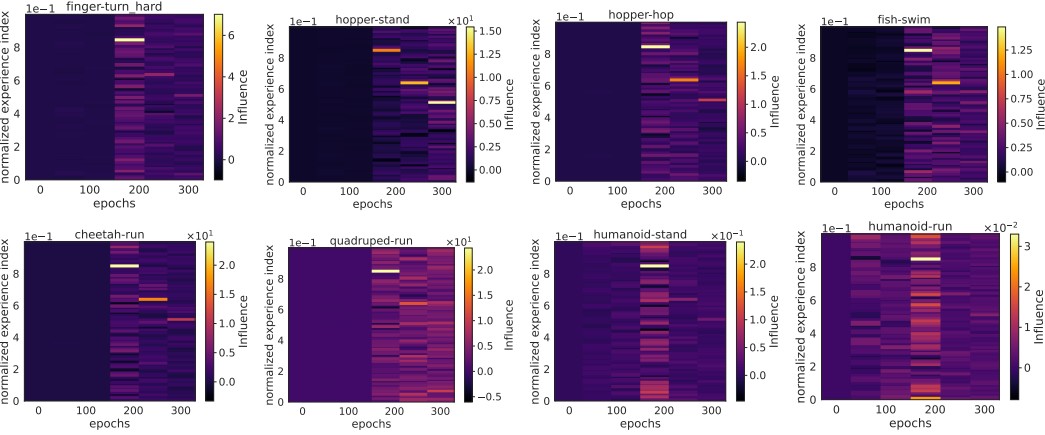

Figure 19: Distribution of influence on Q-estimation bias (Eq. 11) in DM control environments with adversarial experiences.

## G.1 ADDITIONAL EXPERIMENTAL IN DM CONTROL ENVIRONMENTS WITH ADVERSARIAL EXPERIENCES

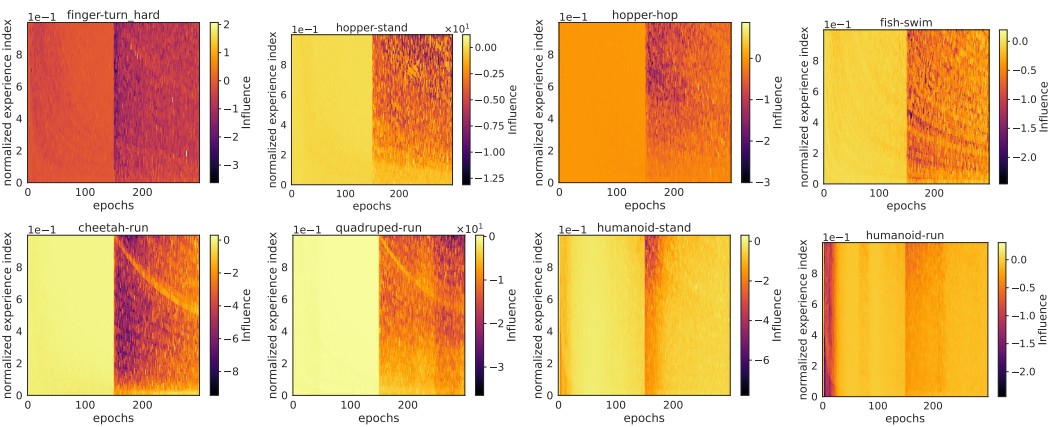

Figure 20: Distribution of influence on policy improvement (Eq. 9) in DM control environments with adversarial experiences.

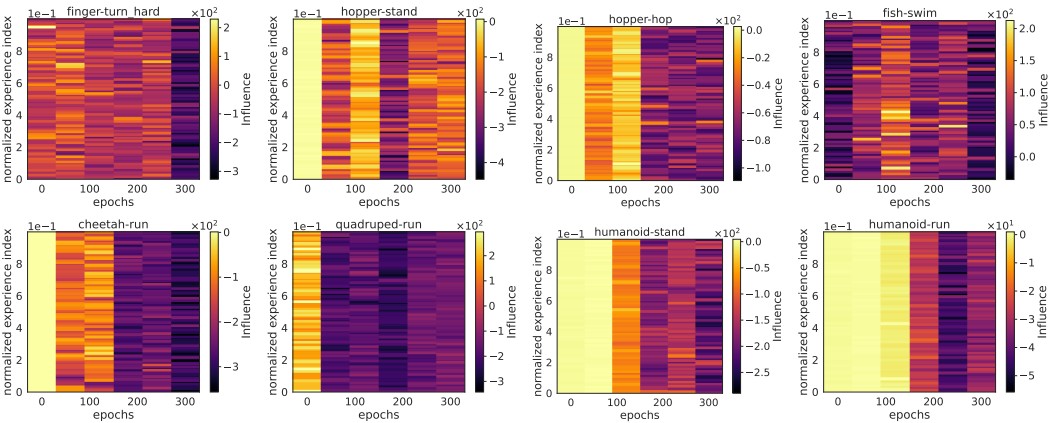

Figure 21: Distribution of influence on return (Eq. 10) in DM control environments with adversarial experiences.

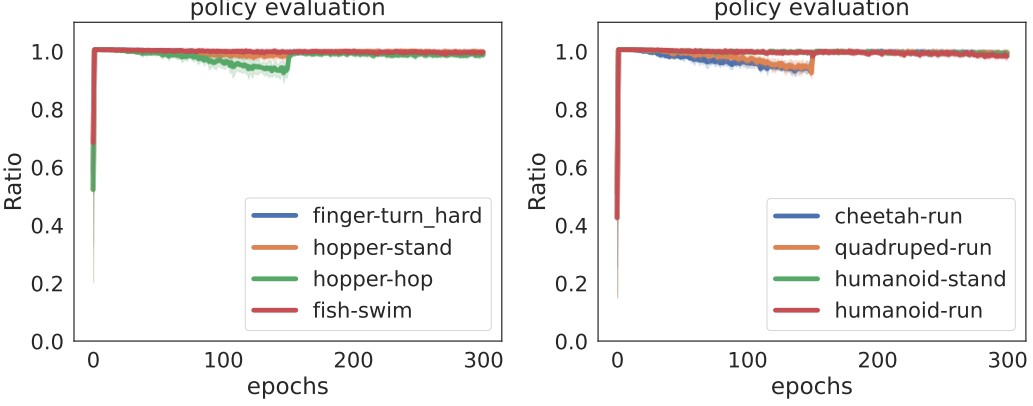

Figure 22: The ratio of experiences for which PIToD correctly estimated influence on policy evaluation (Eq. 8).

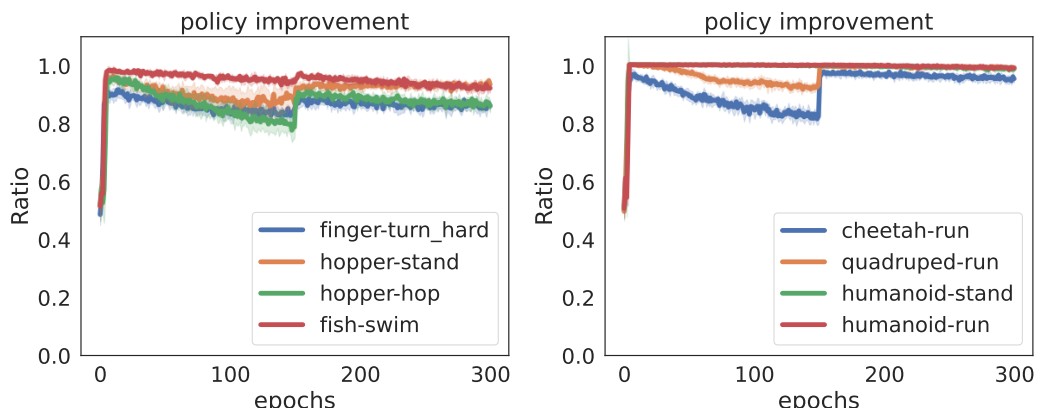

Figure 23: The ratio of experiences for which PIToD correctly estimated influence on policy improvement (Eq. 9).

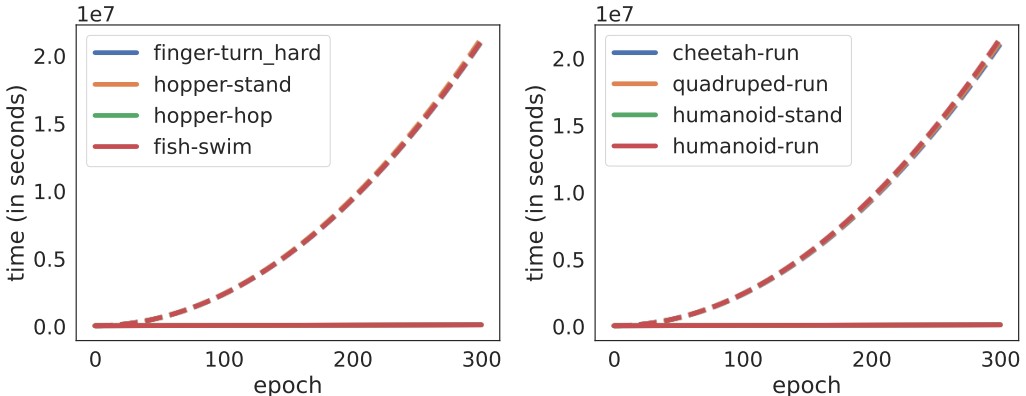

Figure 24: Wall-clock time required for influence estimation by PIToD and LOO. The solid line represents the time for PIToD, and the dashed line represents the estimated time for LOO.

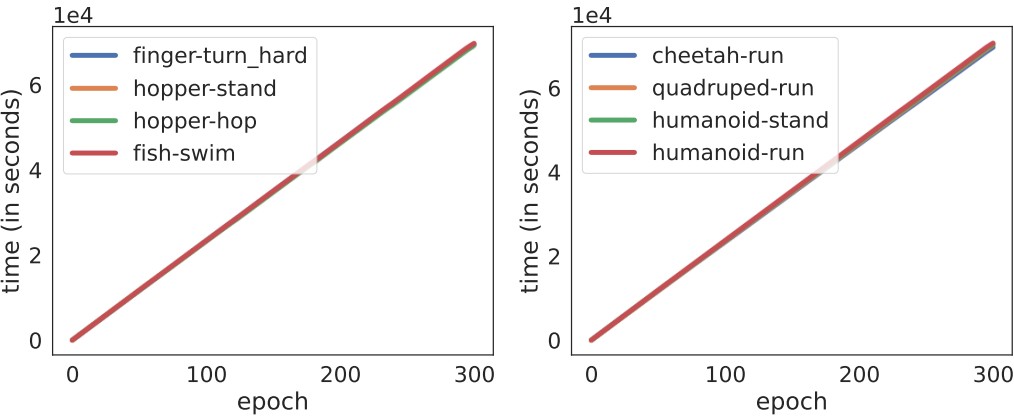

Figure 25: Wall-clock time required for influence estimation by PIToD.

# H AMENDING THE POLICIES AND Q-FUNCTIONS OF DROQ AND RESET AGENTS

In Section 6, we amended the SAC agent using PIToD. In this section, we apply PIToD to amend other RL agents.

We evaluate two PIToD implementations: DroQToD and ResetToD.

**DroQToD** is a PIToD implementation based on DroQ (Hiraoka et al., 2022). DroQ is the SAC variant that applies dropout and layer normalization to the Q-function. DroQToD differs from the original PIToD implementation (Appendix C) in that it has a dropout layer after each weight layer in the Q-function. The dropout rate is set to 0.01 as in Hiraoka et al. (2022). Layer normalization is already included in the Q-function of the original PIToD implementation; thus, no additional changes are made to it.

**ResetToD** is a PIToD implementation based on the periodic reset (Nikishin et al., 2022; D'Oro et al., 2023) of the Q-function and policy parameters. ResetToD differs from the original PIToD implementation in that it resets the parameters of the Q-function and policy every $10^5$ steps.

The policies and Q-functions of these implementations are amended as in Section 6 (i.e., the amendment process follows Algorithm 4 in Appendix D).

The results of the policy and Q-function amendments (Figures 26 and 27) show that the performance of both DroQToD and ResetToD is significantly improved after the amendments. **Return:** For DroQToD, the return is significantly improved after amendment, especially in Hopper (the left side of Figure 26). For ResetToD, the return is significantly improved across all environments (the left side of Figure 27). **Q-estimation bias:** For DroQToD, the estimation bias is significantly reduced after amendment, especially in Humanoid (the right side of Figure 26). For ResetToD, the estimation bias is reduced in the early stages of training (epochs 0–10) in Ant and Walker2d (the right side of Figure 27).

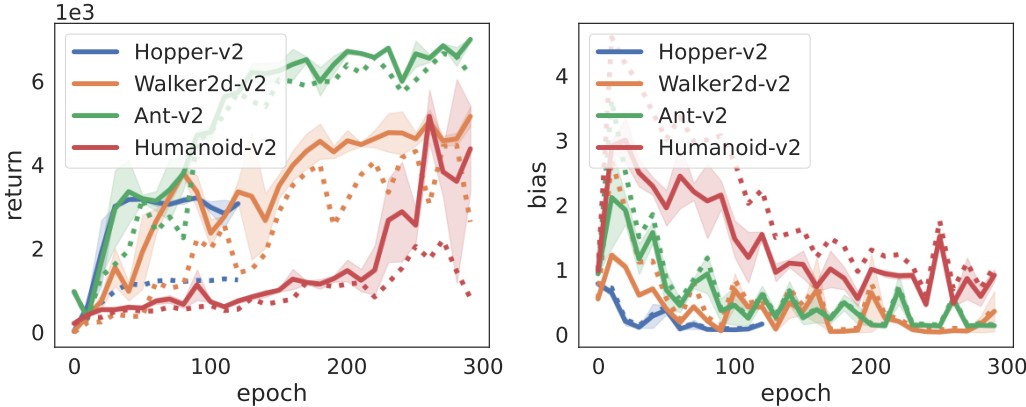

Figure 26: Results of policy amendments (left) and Q-function amendments (right) for DroQToD in underperforming trials. The solid lines represent the post-amendment performances: return for the policy (left; i.e., $L_{\mathrm{ret}}(\pi_{\theta, \mathbf{w}_*})$) and bias for the Q-function (right; i.e., $L_{\mathrm{bias}}(Q_{\phi, \mathbf{w}_*})$). The dashed lines show the pre-amendment performances: return (left; i.e., $L_{\mathrm{ret}}(\pi_\theta)$) and bias (right; i.e., $L_{\mathrm{bias}}(Q_\phi)$).

What experiences negatively influence Q-function or policy performance in the case of DroQToD? Regarding Q-function performance, older experiences negatively influence Q-estimation bias in the early stages of training (the lower part of Figure 31 in Appendix H.1). Regarding policy performance, some experiences negatively influencing returns are associated with wobbly movements. An example of such experiences in the Humanoid environment is shown in the video "DroQToD-Humanoid.mp4," which is included in the supplementary material.

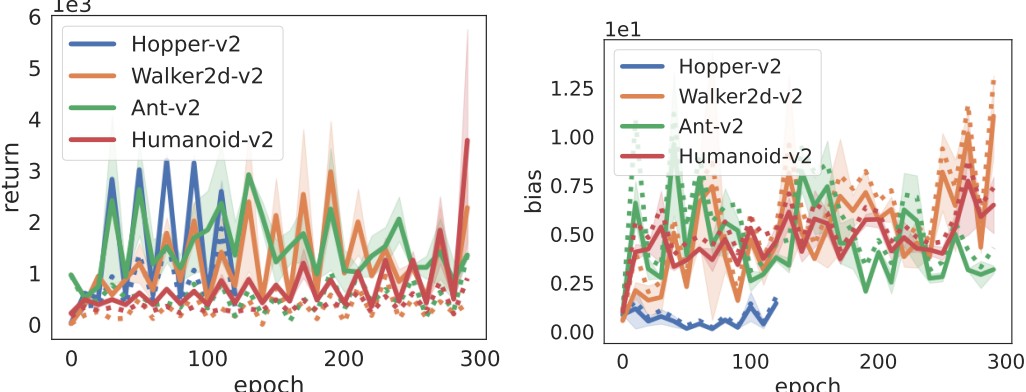

Figure 27: Results of policy amendments (left) and Q-function amendments (right) for ResetToD in underperforming trials. The solid lines represent the post-amendment performances: return for the policy (left; i.e., $L_{\text{ret}}(\pi_{\theta,\mathbf{w}_*})$) and bias for the Q-function (right; i.e., $L_{\text{bias}}(Q_{\phi,\mathbf{w}_*})$). The dashed lines show the pre-amendment performances: return (left; i.e., $L_{\text{ret}}(\pi_\theta)$) and bias (right; i.e., $L_{\text{bias}}(Q_\phi)$).

—

## H.1  ADDITIONAL EXPERIMENTAL RESULTS FOR DROQTOD

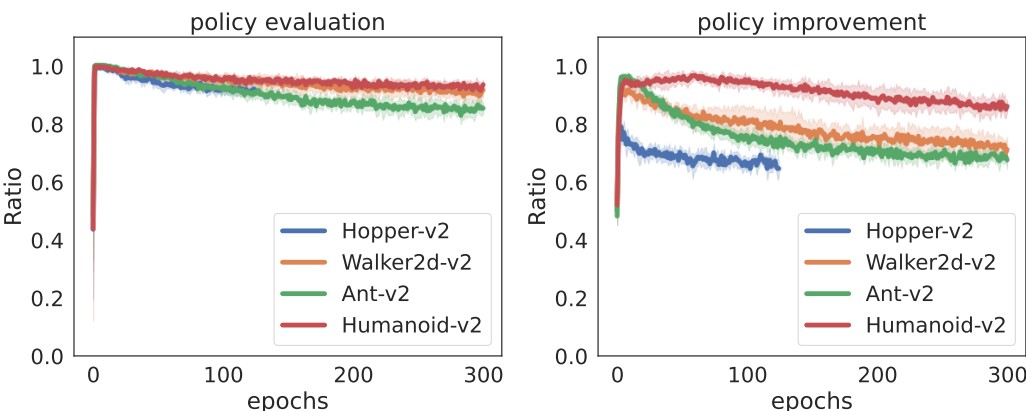

Figure 28: The ratio of experiences for which DroQToD correctly estimated self-influence.

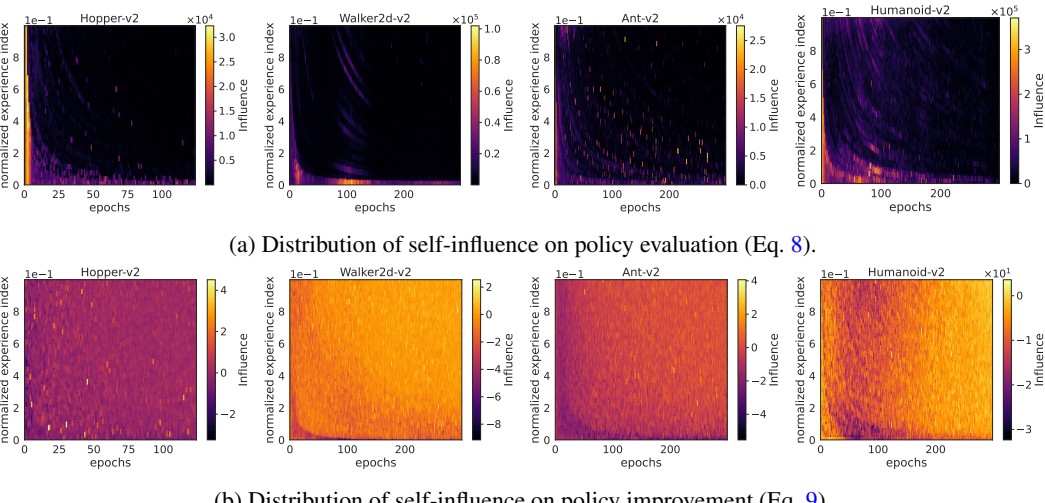

(a) Distribution of self-influence on policy evaluation (Eq. 8).

(b) Distribution of self-influence on policy improvement (Eq. 9).

Figure 29: Distribution of self-influence on policy evaluation and policy improvement.

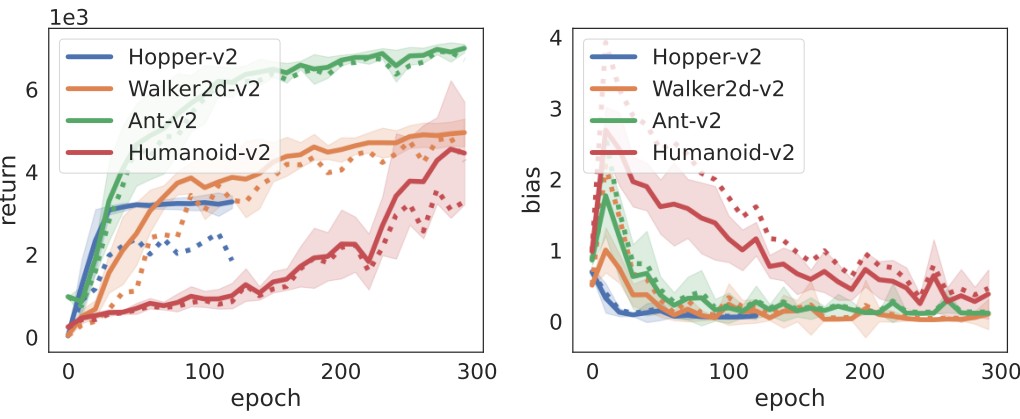

Figure 30: Results of policy amendments (left) and Q-function amendments (right) for all ten trials.

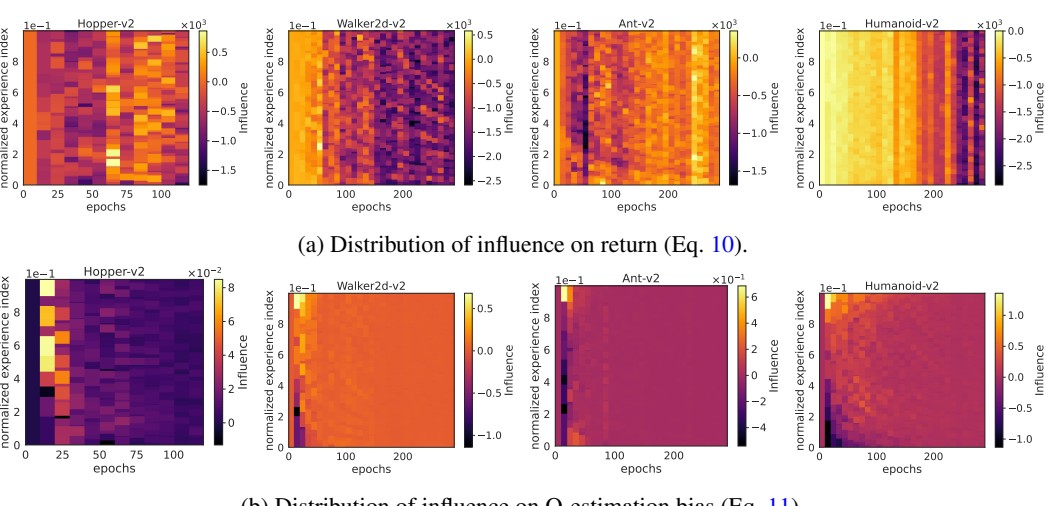

(a) Distribution of influence on return (Eq. 10).

(b) Distribution of influence on Q-estimation bias (Eq. 11).

Figure 31: Distribution of influence on return and Q-estimation bias for all ten trials.

## H.2 ADDITIONAL EXPERIMENTAL RESULTS FOR RESETTOD

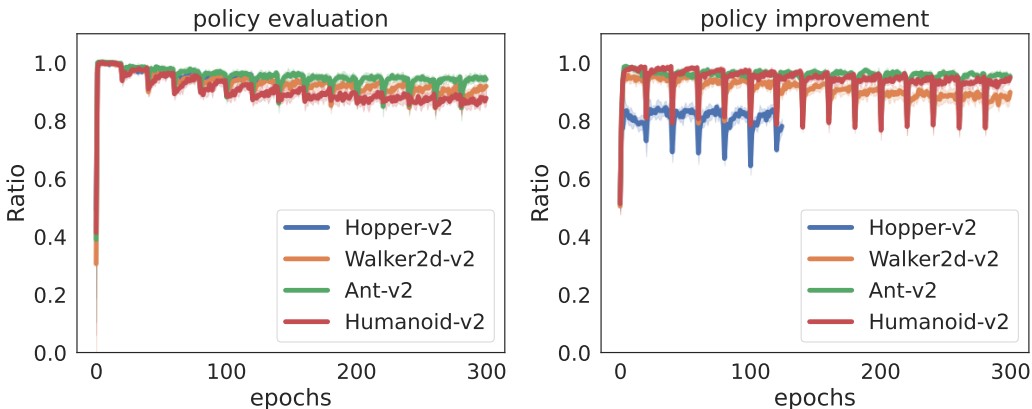

Figure 32: The ratio of experiences for which ResetToD correctly estimated self-influence.

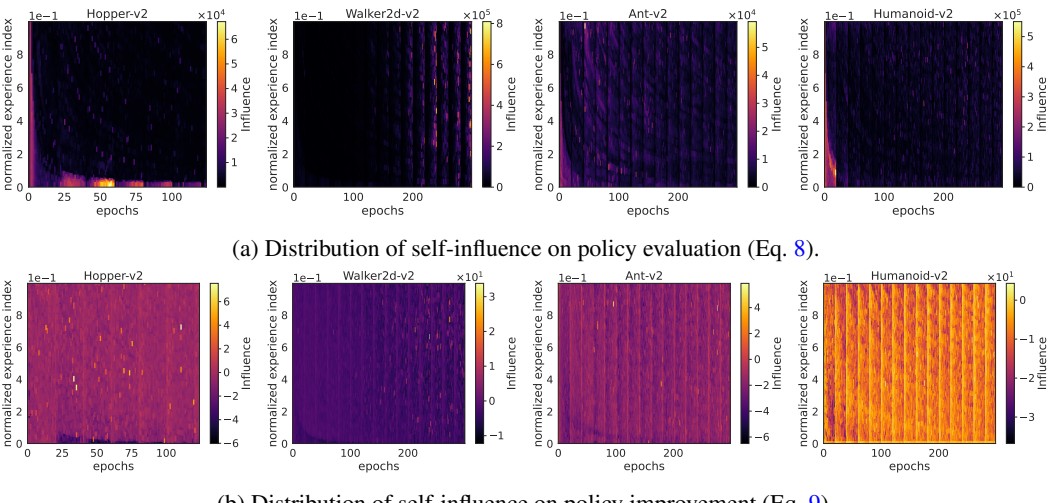

(a) Distribution of self-influence on policy evaluation (Eq. 8).

(b) Distribution of self-influence on policy improvement (Eq. 9).

Figure 33: Distribution of self-influence on policy evaluation and policy improvement.

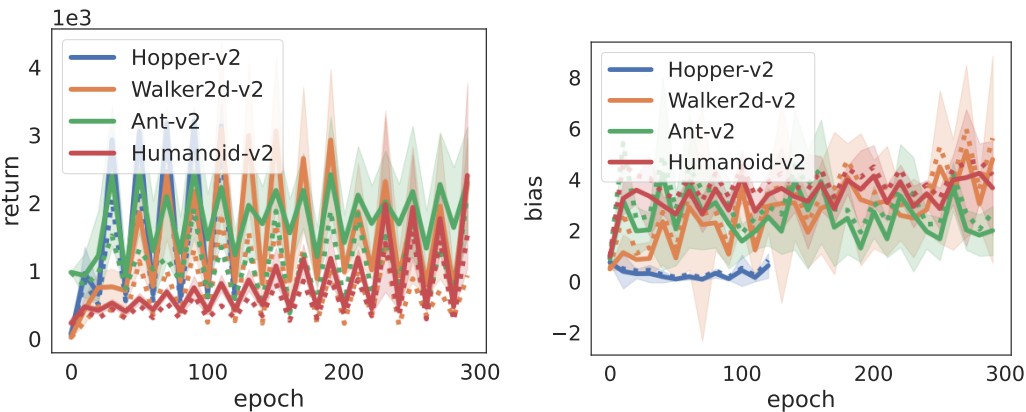

Figure 34: Results of policy amendments (left) and Q-function amendments (right) for all ten trials.

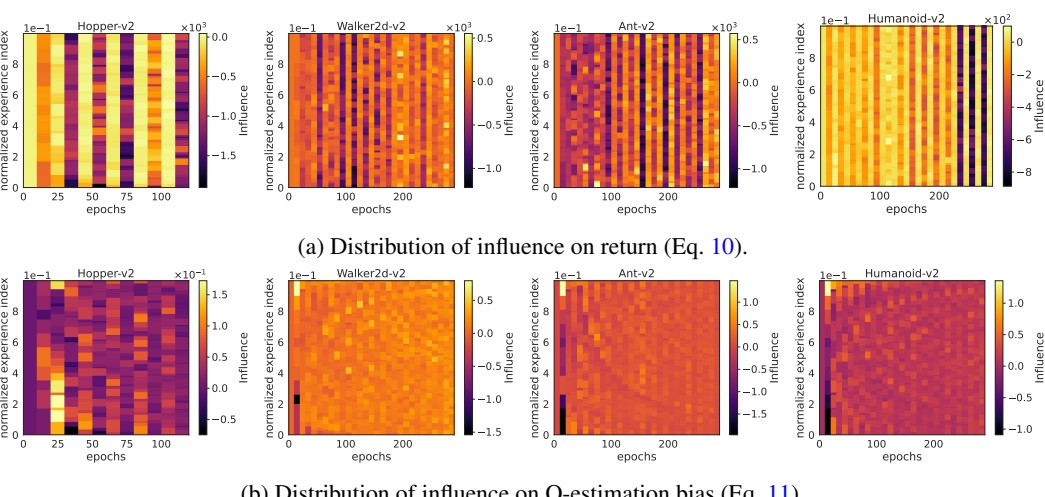

(a) Distribution of influence on return (Eq. 10).

(b) Distribution of influence on Q-estimation bias (Eq. 11).

Figure 35: Distribution of influence on return and Q-estimation bias for all ten trials.

## I LIMITATIONS AND FUTURE WORK

**Refining implementation decisions for PIToD.** PIToD employs a dropout rate of $0.5$ (Section 4 and Appendix B), which often leads to degradation in learning performance. To mitigate this issue, we have considered various design choices in the implementation of PIToD (Appendix C). However, further refinement may still be necessary to improve the practicality of PIToD.

**Overlap of experience masks.** PIToD assigns each experience a randomly generated binary mask (Section 4). When there is significant overlap between the elements of masks, applying the flipped mask to delete the influence of a specific experience also deletes the influence of other experiences. For example, if the masks $\mathbf{m}_i$ and $\mathbf{m}_{i'}$ corresponding to the experiences $e_i$ and $e_{i'}$ have a $100\%$ overlap, applying the flipped mask $\mathbf{w}_i$ completely deletes the influence of both $e_i$ and $e_{i'}$. Additionally, significant overlap between masks may hinder the fulfillment of Assumption 1 and thus compromise the theoretical property derived in Section A. We set the dropout rate of the mask elements to minimize this overlap, but a $50\%$ overlap can still occur (Appendix B). Developing practical methods to reduce mask overlap across experiences would be an important direction for future work.

**Invasiveness of PIToD.** PIToD introduces invasive changes to the base PI method (e.g., DDPG or SAC) to equip it with efficient influence estimation capabilities (Section 4). Specifically, PIToD incorporates turn-over dropout, which may affect the learning outcomes of the base PI method. Consequently, PIToD may not be suitable for estimating the influence of experiences on the original learning outcomes of the base PI method. One direction for future work is to explore non-invasive influence estimation methods.

**Exploring surrogate evaluation metrics for amendments.** To amend RL agents in Section 6, we used the return-based evaluation metric $L_{\mathrm{ret}}$, which requires additional environment interactions for evaluation. In our case, evaluating $L_{\mathrm{ret}}$ required as many as $3 \cdot 10^6$ interactions (Figure 13 in Appendix E). These additional interactions may become a bottleneck in settings where interacting with environments is costly (e.g., real-world or slow simulator environments). Exploring surrogate evaluation metrics that do not require additional interactions is an interesting research direction.

**Exploring broader applications of PIToD.** In this paper, we applied PIToD to amend RL agents in single-task RL settings (Section 6, Appendix G, and Appendix H). However, we believe that the potential applications of PIToD extend beyond single-task RL settings. For instance, it could be applied to multi-task RL (Vithayathil Varghese & Mahmoud, 2020) (including multi-goal RL (Liu et al., 2022) or meta RL (Beck et al., 2023)), continual RL (Khetarpal et al., 2022), safe RL (Gu et al., 2022), offline RL (Levine et al., 2020), or multi-agent RL (Canese et al., 2021). Investigating the broader applicability of PIToD in these settings is a promising direction for future work. Additionally, in this paper, we estimated the influence of experiences by assigning masks to experiences. We may also be able to estimate the influence of specific hyperparameter values by assigning masks to those values. Exploring such applications is another promising direction for future work.

## J  COMPUTATIONAL RESOURCES USED IN EXPERIMENTS

For our experiments in Section 5.2, we used a machine equipped with two Intel Xeon CPUs E5-2667 v4 and five NVIDIA Tesla K80 GPUs. For the experiments in Section G, we used a machine equipped with two Intel Xeon Gold 6148 CPUs and four NVIDIA V100 SXM2 GPUs.

## K  HYPERPARAMETER SETTING

The hyperparameter setting for our experiments (Sections 5 and 6) is described in Table 1. We set different values of $I_{\text{ie}}$ in Sections 5 and 6. In Section 5, we use computationally lighter implementations of evaluation metric $L$ (i.e., $L_{pe,i}$ and $L_{pi,i}$), which allows us to perform influence estimation more frequently; thus, we set a value of 5000 for $I_{\text{ie}}$. On the other hand, in Section 6, we use heavier implementations of $L$ (i.e., $L_{\text{ret}}$ and $L_{\text{bias}}$), and thus set a value of 50000 for $I_{\text{ie}}$.

Table 1: Hyperparameter settings

| Parameter | Value |
|---|---|
| optimizer | Adam (Kingma & Ba, 2015) |
| learning rate | 0.0003 |
| discount rate $\gamma$ | 0.99 |
| target-smoothing coefficient $\rho$ | 0.005 |
| replay buffer size | $2 \cdot 10^6$ |
| number of hidden layers for all networks | 2 |
| number of hidden units per layer | 128 |
| mini-batch size | 256 |
| random starting data | 5000 |
| replay (update-to-data) ratio | 4 |
| masking (dropout) rate | 0.5 |
| influence estimation interval $I_{\text{ie}}$ | 5000 for Section 5 and 50000 for Section 6 |

