# OpenReview forum: "Which Experiences Are Influential for RL Agents? Efficiently Estimating The Influence of Experiences"
_ICLR.cc/2025/Conference — Submitted to ICLR 2025_

### Official Review · Reviewer_A595 · 2024-10-30

**Soundness:** 3
**Presentation:** 3
**Contribution:** 3
**Rating:** 6
**Confidence:** 4

**Summary:**

The paper describes a novel method called Policy Iteration with Turn-over Dropout (PIToD)  for excluding experiences that negatively affect the performance of RL agents when used for training via policy iteration. This includes a how to calculate the influence of a single experience on the agent's performance and how to amend the policy given this calculated influence. This is done efficiently through a parameter masking technique called turn-over dropout. The authors provide theoretical justification as to why this masking technique is similar to leaving out a specific experience. PIToD is tested on known four MuJoCo environments and shows improvement in performance for some of the environments while remaining computationally efficient.

**Strengths:**

- The method is simple, novel, and original.
- Comparison to the leave-one-out naive approach emphasizes the significance of PIToD.
- Diagrams are clear and self-explanatory.
- Results showcase the efficiency advantages of PIToD very well
- the paper is generally well written with a clear narrative and a comfortable flow.

**Weaknesses:**

- the theoretical justification is lacking because the assumptions (1 and 2) don't seem realistic. Basically, the authors assume that the masks enforce some sort of leave-out rule. It makes sense since they try to minimize overlap (App B), but it is the assumptions are unjustified. The authors should provide empirical evidence supporting assumptions 1 and 2, and discuss the implications if these assumptions don't fully hold in practice.
- missing analysis of results, mostly why the results in figures 3, 4 and 6 look the way they do. The authors should explain why the plots for some environments show lower ratios / performance than other, the reasons for instability in some cases (including shaded confidence intervals). Figure 4 is explained but this explanation is unclear. Specifically, it is unclear why this figure suggests that the agent is overfitting to older experiences.
- figure 4 heatmap scales are all different, making it very difficult to read and compare the graphs. Either use consistent scales for all plots or at least keep colors consistent regardless of the scale, i.e., if 1 is yellow and -1 is blue inn one scale, then in another scale -1 is still blue, and -4 can be a new color, e.g., green.
- does not consider what happens if the buffer is full, something that will eventually happen if training persists. The authors should provide a more intuitive explanation or a simple example that illustrates why the signs of these equations indicate correct influence calculations.
- no comparison to other methods. E.g., PER is definitely comparable.
- redundant mini-paragraphs (start of sections) and parentheses (introduction) that map out the content of the paper are distracting and ruin the flow of reading.

**Questions:**

- Could I use PIToD and PER together?
- Your algorithm as many loops. Can these ToD iterations be efficiently batched?
- it is unclear why equations 8 and 9 indicate correct influence calculations if they are positive and negative, respectively. Why is this the case?
- why is the "correct experience ratio" of hopper for in policy improvement so much worse than the others, and why does the ratio for walker and ant decrease throughout the epochs? The authors should discuss potential reasons for these differences and what implications they might have for the applicability of PIToD across different environments.
- why is bias mainly an issue with the humanoid environment and not the others? Is this consistent with previous findings in the literature? What specific characteristics does the humanoid environment have that might contribute to this bias issue?

---

> ### Author Response · Authors · 2024-11-24
> **Response to Reviewer A595**
>
> Thank you for your valuable comments.
> We will revise the paper based on your feedback.
>
> **Q1.**
> the theoretical justification is lacking because the assumptions (1 and 2) don't seem realistic. Basically, the authors assume that the masks enforce some sort of leave-out rule. It makes sense since they try to minimize overlap (App B), but it is the assumptions are unjustified. The authors should provide empirical evidence supporting assumptions 1 and 2, and discuss the implications if these assumptions don't fully hold in practice.
>
> **A1.**
> For Assumption 2, we ensure in our implementation that there is no overlap between the masks $m_i$ and the flip masks $w_i$. Therefore, we believe this assumption holds in practice.
>
> On the other hand, Assumption 1 may not be fully satisfied in our current implementation. If Assumption 1 is not strictly met, it means that applying the flip mask corresponding to a specific experience may not entirely exclude the influence of that experience. To fully satisfy Assumption 1, one potential implementation would involve completely eliminating any overlap between the masks $m_i$ and $m_j$ ($i \neq j $). However, this approach would require ignoring interactions between experience groups, which is impractical. For this reason, we did not adopt such a method in our implementation.
> In summary, there is a trade-off between strictly meeting Assumption 1 and maintaining practicality. Research and development on methods and implementations that balance these factors remain future work.
> We will add a discussion of this point in Appendix I.
>
>
>
>
> **Q2.**
> missing analysis of results, mostly why the results in figures 3, 4 and 6 look the way they do. The authors should explain why the plots for some environments show lower ratios / performance than other, the reasons for instability in some cases (including shaded confidence intervals). Figure 4 is explained but this explanation is unclear. Specifically, it is unclear why this figure suggests that the agent is overfitting to older experiences.
>
> **A2.**
> Regarding the explanation of Figure 4, the phrase "overfitting to older experiences" may have been an overstatement. We have revised the manuscript to clarify that self-influence is concentrated on older experiences, without implying overfitting.
> For the other points (Figures 3 and 6) you mentioned, we will include additional explanations and analyses in the revised manuscript.
>
>
> **Q3.**
> figure 4 heatmap scales are all different, making it very difficult to read and compare the graphs. Either use consistent scales for all plots or at least keep colors consistent regardless of the scale, i.e., if 1 is yellow and -1 is blue inn one scale, then in another scale -1 is still blue, and -4 can be a new color, e.g., green.
>
> **A3.**
> We are currently revising the figure to address this issue.
>
>
> **Q4.**
> does not consider what happens if the buffer is full, something that will eventually happen if training persists.
>
> **A4.**
> If computational resources allow, we will conduct experiments to address this scenario.
>
> \# In PIToD, even if the buffer becomes full and older experiences are overwritten by newer ones, this does not pose a fundamental problem. However, if we want to estimate the influence of the overwritten experiences later, it is necessary to record the corresponding masks (or the random seeds used to generate those masks).
>
>
>
>
> **Q5.**
> no comparison to other methods. E.g., PER is definitely comparable.
>
> **A5.**
> We are currently conducting experiments with SAC. Once those are complete, we will add experiments with SAC+PER as well.
>
> **Q6.**
> Could I use PIToD and PER together?
>
> **A6.**
> PER performs weighted sampling of experiences based on TD errors evaluated with the Q-function ([1], Algorithm 1, line 11).
> If it is acceptable to ignore the influence of individual experiences on the sampling process, then PIToD and PER can be used together.
> In such a case, combining the two methods would involve replacing PIToD's uniform sampling scheme with PER's prioritized sampling scheme.
>
> [1] https://arxiv.org/pdf/1511.05952
>
>
> **Q7.**
> Your algorithm as many loops. Can these ToD iterations be efficiently batched?
>
> **A7.**
> For line 8 of Algorithm 2, when using $L_{pe, i} $, $L_{pi, i}$, and $L_{bias, i}$, the operations are performed in batches.
> These batch operations are implemented in the provided code under `redq/utils/bias_utils.py` in the supplemental material.
> $L_{ret}$ involves interaction with the Mujoco (CPU) environment for return calculation, so batching is not applied in this case.

---

> > ### Author Response · Authors · 2024-11-24
> > **Response to Reviewer A595**
> >
> > **Q8.**
> > The authors should provide a more intuitive explanation or a simple example that illustrates why the signs of these equations indicate correct influence calculations.
> > **Q8'.**
> > it is unclear why equations 8 and 9 indicate correct influence calculations if they are positive and negative, respectively. Why is this the case?
> >
> > **A8.**
> > The explanation for the reason is provided in the second paragraph of Section 5.1:  "We evaluate whether PIToD has correctly estimated the influence of experiences by examining the signs (positive or negative) of the values of Eq. 8 and Eq. 9...."
> >
> > The policy and Q-function trained with the mask are optimized to maximize $L_{pi, i}$ and minimize $L_{pe, i}$, respectively (c.f. lines 5 and 6 in Algorithm 2).
> > In contrast, the policy and Q-function trained with the flip mask are not optimized in this way. Therefore:
> > - The sign of Eq. 8 is positive because the mask reduces the TD error more significantly.
> > - The sign of Eq. 9 is negative because the mask results in higher action values.
> >
> >
> > **Q9.**
> > why is the "correct experience ratio" of hopper for in policy improvement so much worse than the others, and why does the ratio for walker and ant decrease throughout the epochs? The authors should discuss potential reasons for these differences and what implications they might have for the applicability of PIToD across different environments.
> >
> > **A9.**
> > For policy improvement, it appears that environments with higher-dimensional action and state spaces tend to have higher ratios (Hopper < Walker == Ant < Humanoid). This suggests that higher-dimensional spaces might allow the policy function to better discriminate between experiences.
> >
> > As for the decreasing ratio in some environments, this could be due to an increasing proportion of similar experiences in the replay buffer as epochs progress, or an increase in the variety of experiences relative to the network (mask) size, making it harder to differentiate individual experiences.
> >
> > We have added this discussion to the main text.
> >
> >
> >
> >
> > **Q10.**
> > why is bias mainly an issue with the humanoid environment and not the others? Is this consistent with previous findings in the literature? What specific characteristics does the humanoid environment have that might contribute to this bias issue?
> >
> > **A10.**
> > In the Humanoid environment, Q-functions are more prone to overestimation, making bias a more significant issue.
> > This trend is consistent with findings in previous research (e.g., Figure 14 in [1]).
> > Generally, tasks with higher-dimensional state and action spaces tend to exhibit this type of problem more prominently.
> >
> > [1] https://arxiv.org/pdf/2110.02034

---

> > > ### Comment · Reviewer_A595 · 2024-11-25
> > >
> > > A1. the justification for assumption 2 seems valid. assumption 1 still seems unrealistic and a more in-depth analysis of the aforementioned tradeoff is required. furthermore, no discussion was added to the appendix.
> > >
> > > A2. The added explanations are unclear:
> > >     fig 3: what does it mean to "fit more significantly" to each experience? can you provide an explanation as to why this is happening, other than "high dimensions"?
> > >     fig 4: (a) It is still unclear why these phenomena happen and no explanation / hypothesis is provided. (b) if there is no pattern then what is the significance of these results?
> > >     fig 6: No additional explanation was found. was any new information on this figure added to the text? if so, where?
> > >
> > > A3. the figure still requires fixing
> > >
> > > A4. Though filling up the buffer does not pose a fundamental problem in running PIToD, it is still a concern that new, bad experiences can replace old, good experiences. can PIToD be used to help choose which experiences to discard? further analysis is required.
> > >
> > > A5. experiment results were not yet added to the paper
> > >
> > > A6. what is the meaning of "ignore the influence of individual experiences on the sampling process"? isn't that the main contribution for PIToD?
> > >
> > > A7. very interesting. thank you for your response
> > >
> > > A8. this explanation simply states how the network was optimized, but it is still unclear what "correct" means in this context and why this correctness corresponds to the sign of these equations.
> > >
> > > A9. What is the intuition behind this explanation? why does this suggest that "higher-dimensional spaces might allow...". The explanation about the decreasing ratio makes sense.
> > >
> > > A10. This is an important point to understand when reading this plot. please add this citation with a small explanation in the main text (preferably direct the reader to figure 14 directly)

---

> > > > ### Author Response · Authors · 2024-12-01
> > > >
> > > > Thank you for your comments.
> > > > Due to the limited time available for the rebuttal, we may not be able to address all your comments, but we are working on incorporating as many of them as possible.
> > > >
> > > >
> > > > Below are responses to some of your questions:
> > > >
> > > > > A2. The added explanations are unclear: fig 3: what does it mean to "fit more significantly" to each experience? ...
> > > >
> > > > By "fit," we mean that applying the mask allows (i) the policy to achieve higher estimated action values and (ii) the Q-function to reduce the TD error, compared to applying the flipped mask.
> > > > Rather than explaining this by using Fig. 3, it would be clearer to explain this by using the histogram of self-influence values. We will add this explanation and figure of the histogram.
> > > >
> > > >
> > > > > A6. What is the meaning of "ignore the influence of individual experiences on the sampling process"? Isn't that the main contribution of PIToD?
> > > >
> > > > PIToD assumes that experience sampling from the replay buffer does not depend on experiences (e.g., uniform sampling). Under this assumption, PIToD efficiently tracks the influence of experiences during policy iteration.
> > > > Therefore, when sampling is influenced by experiences (e.g., as in PER), it is necessary to either conduct additional discussions to address this or disregard influences on the experience sampling part.
> > > >
> > > >
> > > >
> > > > We sincerely appreciate your detailed comments and your dedicated engagement during the author rebuttal period.

---

### Official Review · Reviewer_YodU · 2024-10-30

**Soundness:** 2
**Presentation:** 2
**Contribution:** 2
**Rating:** 3
**Confidence:** 3

**Summary:**

This paper presents Policy Iteration with Turn-over Dropout as a method for estimating the affect a state-action-reward-next state experience has on a policy or Q function in the area of off-policy RL. The method employs masks to dropout parameters, so that the affect of not training on an experience can be estimated without having to retrain the policy or Q values from scratch. The paper then investigates estimating various quantities including td-error and episode return.

**Strengths:**

Being able to efficiently estimate the affect a particular data point has on a network used to estimate $\pi$ or $Q$ is a very powerful tool, and to my knowledge this machinery has not been applied to a Deep RL setting before. I like that the authors have aimed to evaluate it for a variety of different purposes.
However, it seems that only Section G in the appendix, and a short paragraph at the end of section 6 actually attempt to answer the question in the title of the paper.

**Weaknesses:**

# Major

More explanation in Section 4 of the PIToD method would be very helpful for a reader's understanding.
"Thus, some readers may suspect that the parameters dropped out by $m_i$ (i.e., the parameters obtained by applying $w_i$) are not influenced by $e_i$." - This sentence caused a lot of confusion to me when reading the paper. The phrasing seems to suggest that the parameters dropped out are indeed affected, but this doesn't seem to be the case (as the reader would suspect). I am also struggling to understand Appendix Section A. Assumption 1 seems extremely strong and appears to be *almost* exactly defining the property you are looking to prove. Looking at the equality, doesn't it mean that the gradient for $i'$ is 0 for everything but $i'$ due to the indicator?

Section C in the appendix contains a lot of interesting content.
The findings from the Group Mask preliminary experiments in the appendix seem quite important - "In our preliminary experiments, we found that the influence of a single experience on performance was negligibly small". This should be mentioned at the very least in the main paper.
"Key implementation decisions to improve learning."  is also very important. The implementation details, and the architectural experiments you conducted should not be relegated to the appendix in this manner. Especially since Figure 10 shows they are critical to your method working.
Additionally, why are you using an ensemble of 20 MLPs in the architecture in this manner? Is it important to performance? Does it interact positively with your method? More discussion on why these design decisions were made is needed, or at the very least a comment stating it was the first architectural starting point used.

More explanation is needed on Section 5.1 to establish the importance of the ratios being considered. What is the importance/significance of the differences in the ratios between policy eval/improvement? For Eqn 8 I can understand that the td error if higher for an experience you haven't trained on, but for Eqn 9 I do not quite see why the action chosen by $\pi_{\theta, w_i}$ should have a lower estimated Q.

Section 5.2 is very misleading since you're estimating the time it would take LOO. Given that important sections that have been pushed into the appendix, I would advise replacing these with other more relevant/concrete content.

"In our setup, L_ret is estimated using Monte Carlo returns collected by rolling out policies" - How many rollouts are used for each estimation?
Additionally, how do you utilise Eq. 10 to identify experiences? Are you rolling out the agent multiple times to collect MC returns? Is this factored into your training budget and reflected in Figure 6?

I don't understand why you're not showing Figure 11 in place of Figure 6? What is the particular thing you want to highlight by showing a specially picked subset of the lines in Figure 6?

Much more analysis and results need to be presented on characterising which experiences are harmful (or beneficial). The paper begins this kind of investigation but it feels like an afterthought. To me, this is one of the most exciting parts of the paper, utilising the tools outlined to identify what experiences are harmful for performance (with potential links to 'Ray Interference: a Source of Plateaus in Deep Reinforcement Learning'). This would be of much interest to the community.

# Minor

- No references for RL, and an MDP is never mentioned in the paper.
- No reference for the LOO estimator.
- CQL cited under off-policy RL in the introduction feels unnecessary.
- "In the previous section, we demonstrated that PIToD can accurately and efficiently estimate the influence of experiences." - This is too broad a claim. In section 5 you estimated the influence an experience can have on self-influence.

**Questions:**

- For section G, Figure 16 shows very little change in the results when removing adversarial experiences, why is this? Figure 17 shows big differences in the estimations before and after amendments, but it would be good to clarify that your method can identify the adversarial experiences explicitly (as opposed to showing the affect of removing identified experiences which indirectly provides some evidence for this).

- Figure 10 shows huge changes in the results across some architectural choices, please comment more on these.

(There are also some questions sprinkled throughout the above section)

---

> ### Author Response · Authors · 2024-11-24
> **Response to Reviewer YodU**
>
> Thank you for your valuable comments.
> We will revise the paper based on your feedback.
>
> **Q1.**
> More explanation in Section 4 of the PIToD method would be very helpful for a reader's understanding. "Thus, some readers may suspect that the parameters dropped out by  (i.e., the parameters obtained by applying ) are not influenced by ." - This sentence caused a lot of confusion to me when reading the paper. The phrasing seems to suggest that the parameters dropped out are indeed affected, but this doesn't seem to be the case (as the reader would suspect).
>
> **A1.**
> We intended to use "suspect" in the sense of "think" or "believe," rather than "doubt," which may have caused some confusion.
> The parameters dropped out by the mask are, in theory, not influenced by the corresponding experience.
> We have revised the sentence in Section 4 to ensure that this point is communicated clearly and unambiguously. Thank you for highlighting this potential misunderstanding.
>
>
> **Q2.**
> I am also struggling to understand Appendix Section A....  Looking at the equality, doesn't it mean that the gradient for $i'$ is 0 for everything but $i'$ due to the indicator?
>
> **A2.**
>
> > Assumption 1 seems extremely strong and appears to be almost exactly defining the property you are looking to prove.
>
> Assumption 1 may be a strong assumption that is difficult to reconcile with practical implementations.
> As mentioned in response A1 to Reviewer A595, strictly satisfying this assumption in practical implementations is not easy at this stage.  i.e., Prioritizing practicality means that fully meeting this assumption remains challenging, and resolving this gap is a topic for future work.
>
> Assumption 1 essentially assumes that the Q-function and policy function using $w_i'$ can be replaced by functions dominantly influenced by the experience $e_i'$.
> On the other hand, what we aim to prove is that, when training under the PIToD framework, applying the flip mask $w_i'$ corresponding to $e_i$ enables us to isolate parameters unaffected by $e_i$.
>
> > Looking at the equality, doesn't it mean that the gradient for is 0 for everything but due to the indicator?
>
> Yes, that is correct.
>
>
> **Q3.**
> Section C in the appendix contains a lot of interesting content. The findings from the Group Mask preliminary experiments in the appendix seem quite important - "In our preliminary experiments, we found that the influence of a single experience on performance was negligibly small". This should be mentioned at the very least in the main paper. "Key implementation decisions to improve learning." is also very important. The implementation details, and the architectural experiments you conducted should not be relegated to the appendix in this manner. Especially since Figure 10 shows they are critical to your method working.
>
> **A3.**
> As per your suggestion, we plan to include several key insights and findings in the main paper.
>
>
> **Q4.**
> What is the importance/significance of the differences in the ratios between policy eval/improvement? For Eqn 8 I can understand that the td error if higher for an experience you haven't trained on, but for Eqn 9 I do not quite see why the action chosen by $\pi_{\theta, w_i}$ should have a lower estimated Q.
>
> **A4.**
> $\pi_{\theta, m_i}$ is trained to maximize the estimated Q (c.f. Algorithm 2, line 6), whereas $\pi_{\theta, w_i}$ is not trained in the same way.
> As a result, $\pi_{\theta, w_i}$ is expected to have a lower estimated Q.
>
>
> **Q5.**
> Section 5.2 is very misleading since you're estimating the time it would take LOO. Given that important sections that have been pushed into the appendix, I would advise replacing these with other more relevant/concrete content.
>
> **A5.**
> Thank you for pointing this out. To address your concern, we are considering replacing this section with other (e.g., Appendix C).
> Additionally, we will explicitly clarify in the text that the LOO time presented in this section is based on an estimated calculation.
>
>
>
> **Q6.**
> "In our setup, $L_{ret}$ is estimated using Monte Carlo returns collected by rolling out policies" - How many rollouts are used for each estimation? Additionally, how do you utilise Eq. 10 to identify experiences? Are you rolling out the agent multiple times to collect MC returns? Is this factored into your training budget and reflected in Figure 6?
>
> **A6.**
> For each group of experiences included in the replay buffer, the agent is rolled out for 10 test episodes to estimate the MC return. Using these estimates, Eq. 10 is calculated to estimate the influence of each experience group.
>
> The experience group corresponding to the flip mask with the highest Eq. 10 value is identified as having a negative influence.
>
> The additional budget required for influence estimation is not reflected in Figure 6 but is shown in Figure 13.

---

> > ### Author Response · Authors · 2024-11-24
> > **Response to Reviewer YodU**
> >
> > **Q7.**
> > I don't understand why you're not showing Figure 11 in place of Figure 6? What is the particular thing you want to highlight by showing a specially picked subset of the lines in Figure 6?
> >
> > **A7.**
> > This is because we are particularly interested in the removal of negatively influential experiences in cases where learning does not proceed well.
> > Figure 11 shows the results of removal in the average case, which includes many trials where learning is proceeding without issues, and removing negatively influential experiences has little effect.
> >
> >
> > **Q8.**
> > Much more analysis and results need to be presented on characterising which experiences are harmful (or beneficial). The paper begins this kind of investigation but it feels like an afterthought. To me, this is one of the most exciting parts of the paper, utilising the tools outlined to identify what experiences are harmful for performance (with potential links to 'Ray Interference: a Source of Plateaus in Deep Reinforcement Learning'). This would be of much interest to the community.
> >
> > **A8.**
> > We plan to conduct additional analysis on harmful and beneficial experiences and include these findings in the paper.
> > Thank you for the reference; we will also explore its relevance and include a discussion if possible.
> >
> >
> >
> > **Q9.**
> > Minor comments
> >
> > **A9.**
> > Thank you for your feedback. We will make the revisions as you pointed out.
> >
> >
> >
> > **Q10.**
> > For section G, Figure 16 shows very little change in the results when removing adversarial experiences, why is this? Figure 17 shows big differences in the estimations before and after amendments, but it would be good to clarify that your method can identify the adversarial experiences explicitly (as opposed to showing the affect of removing identified experiences which indirectly provides some evidence for this).
> >
> > **A10.**
> > The little change observed in Figure 16 is likely due to overlap between the masks of different experiences.
> >
> > The ability of our method to explicitly identify adversarial experiences is discussed in the final paragraph of Appendix G ("Can PIToD identify adversarial experiences?...").
> >
> >
> > **Q11.**
> > Additionally, why are you using an ensemble of 20 MLPs in the architecture in this manner? Is it important to performance? Does it interact positively with your method? More discussion on why these design decisions were made is needed, or at the very least a comment stating it was the first architectural starting point used.
> > **Q11'.** Figure 10 shows huge changes in the results across some architectural choices, please comment more on these.
> >
> > **A11.**
> > We initially tried directly applying dropout to individual parameters, but this approach was too unstable.
> > To address this, we explored 50 to 100 alternative methods for stabilizing training. Among these, the ensemble-based approach we used was the most effective. It is crucial for ensuring the performance of PIToD.
> > Typically, an ensemble size of 5 is sufficient for improving performance. However, we observed that increasing the ensemble size tends to make it easier to estimate the influence of experiences. Based on this observation and our computational resources, we chose an ensemble size of 20.

---

> > > ### Comment · Reviewer_YodU · 2024-11-29
> > > **Thanks for your reply**
> > >
> > > Based on your replies to mine and other reviews I am inclined to keep my score.
> > > There's a lot of feedback and suggestions in the reviews that can be used to strengthen the paper.
> > >
> > > 1. The overly strong assumptions make the contribution of the theory much weaker, and these should be highlighted more strongly in the paper so the reader is aware of the disconnect between a practical implementation and the theoretical results.
> > > 2. Please do make clear the additional budget used for estimating the MC returns, and include it in the graphs so a reader can easily tell how many environmental experiences your method is using.
> > > 3. "This is because we are particularly interested in the removal of negatively influential experiences in cases where learning does not proceed well." - make this clear in the main paper. Especially that the results look quite different in the average case.
> > > 4. "To address this, we explored 50 to 100 alternative methods for stabilizing training." - I realise that you cannot possibly include all of these, but practitioners in this area would benefit from reading about your specific experiences with what worked/what didn't work. To me, it seems like the more empirical side of the paper has a lot of work that is not presented/not highlighted which is a shame since its very important to getting the method to work well.

---

> > > > ### Author Response · Authors · 2024-12-01
> > > >
> > > > Thank you for your comments and your engagement during the author rebuttal period.
> > > > We will do our best to incorporate your suggestions into the next revision.

---

### Official Review · Reviewer_cTe3 · 2024-11-01

**Soundness:** 1
**Presentation:** 1
**Contribution:** 2
**Rating:** 3
**Confidence:** 3

**Summary:**

This paper proposes the PIToD method to efficiently estimate which experiences positively or negatively impact RL learning. The proposed approach sets a drop-out mask for each experience and estimates the influence of each experience based on this mask and its complement, allowing for significantly more efficient computation compared to traditional Leave-One-Out (LOO) methods. The authors demonstrate that their method accurately estimates the influence of experiences using various metrics and further show, through experiments, that applying it to SAC improves learning performance.

**Strengths:**

1. A novel approach is presented for estimating the influence of specific experiences in RL by using Turn-over Dropout (ToD).
2. It is theoretically demonstrated that the complement mask $w\_i$ for experience $e\_i$ indicates an absence of influence from $e\_i$.

**Weaknesses:**

1. The metric used to show self-influence appears inappropriate. As noted in L246,  $Q\_{w\_i}$ is in a state where it has not been trained on $e\_i$, so $L\_{pe,i}(Q\_{w\_i}) − L\_{pe,i}(Q\_{m\_i})$ is expected to be greater than zero in most cases, regardless of whether $e\_i$ is beneficial for learning. Additionally, since $\pi\_{m\_i} = \arg\max\_{\pi} L\_{pi,i}(\pi)$, $L\_{pi,i}(\pi\_{w\_i}) − L\_{pi,i}(\pi\_{m\_i})$ is likely always less than zero. Thus, these metrics do not directly indicate whether the experience has a positive or negative influence on RL learning.
2. The evaluation shown in Figure 6 also seems flawed. In Algorithm 4 of Appendix D,  $w^*$ is already defined as  $\arg\max\_w L\_{ret}(\pi, w)$ , so it is unsurprising that high returns are achieved.
3. The main paper contains too few experimental results. While it seems that several experiments were conducted in the appendix, summarizing the purpose and outcomes of these experiments in the main paper would enhance clarity.
4. It would be beneficial to include a direct comparison of mean performance between the original SAC method and the SAC method with PIToD in the main paper.

**Questions:**

1. **[About Weakness 1]** You mentioned that the PI and PE metrics indicate whether a specific experience has a positive or negative effect. Could you clarify this further? The current metrics seem to only reflect whether or not learning utilized $e_i$.
2. **[About Weakness 2]** Is this approach fundamentally different from simply creating a policy ensemble through dropout and selecting the best-performing one? If the truly optimal  $w^*$  is selected, does it necessarily mean that the experience has a negative impact?
3. **[About Weakness 3]** Could you summarize the experiments in the appendix and explain what each aims to demonstrate? It would be more beneficial if these results were integrated into the main paper.
4. **[About Weakness 4]** Could you also present the experiments mentioned above?

---

> ### Author Response · Authors · 2024-11-24
> **Response to Reviewer cTe3**
>
> Thank you for your valuable comments.
> We will revise the paper based on your feedback.
>
> **Q1.**
> The metric used to show self-influence appears inappropriate. As noted in L246,  is in a state where it has not been trained on , so  is expected to be greater than zero in most cases, regardless of whether  is beneficial for learning. Additionally, since , is likely always less than zero. Thus, these metrics do not directly indicate whether the experience has a positive or negative influence on RL learning.
> **Q1'.**
> [About Weakness 1] You mentioned that the PI and PE metrics indicate whether a specific experience has a positive or negative effect. Could you clarify this further? The current metrics seem to only reflect whether or not learning utilized $e_i$.
>
> **A1.**
> Regarding the PI and PE metrics in Section 5.1, we did not claim that "the PI and PE metrics indicate whether a specific experience has a positive or negative effect."
> It seems there might be a difference in interpretation of the terms "positive" and "negative."
> In Section 5.1, we use "positive" to mean a value greater than or equal to zero and "negative" to mean a value less than or equal to zero.
> We are not using "positive" to imply that an experience is beneficial for learning or "negative" to imply that it is harmful.
>
>
> **Q2.**
> [About Weakness 2] Is this approach fundamentally different from simply creating a policy ensemble through dropout and selecting the best-performing one? If the truly optimal is selected, does it necessarily mean that the experience has a negative impact?
>
> **A2.**
> This approach fundamentally differs from a simple policy ensemble in that specific policies within the ensemble are trained exclusively on specific sets of experiences.
> If the optimal policy has not been trained on a particular set of experiences and outperforms those that have, it indicates that the specific set of experiences has a negative impact.
>
>
> **Q3.**
> [About Weakness 3] Could you summarize the experiments in the appendix and explain what each aims to demonstrate? It would be more beneficial if these results were integrated into the main paper.
>
> **A3.**
> We will add a summary of the objectives and results of the experiments in the appendix to the main paper.
>
>
> **Q4.**
> [About Weakness 4] Could you also present the experiments mentioned above?
>
> **A4.**
> We are currently conducting comparison experiments with SAC, and we will add the results once they are complete.

---

### Official Review · Reviewer_xR4M · 2024-11-07

**Soundness:** 3
**Presentation:** 2
**Contribution:** 1
**Rating:** 3
**Confidence:** 3

**Summary:**

This paper aims to study the influence individual experience sample have in training RL agents. The authors used identify masks to differentiate individual samples and observe the difference in the resulting Q values.

**Strengths:**

- This paper focus on experience replay, that is, sampling distribution manipulation, which I think is a under-represented direction in RL research.
- I like the the fact that ToD being applied in the sample consideration, the usage of ToD feels natural and justified for this use case.

**Weaknesses:**

- The paper do not have a theoretical underpinning for their approach though some reader may find the idea intuitive, that said, personally, I'm not a fan of removing samples from experience buffers;
- because the way I see it, deleting "negatively influential experiences" seems to be a lenient/hysteresis update commonly seen in optimistic approaches, it may be useful in same case but should be used with caution since it may hinder learning and cause bias; a comparison with similar approaches is not included. I think instead of removing samples, the safer approach is to use gradient clipping or learning rate schedules to stabilize training. Techniques like gradient clipping cap the maximum value of gradients to prevent instability without losing any data --
- Data removal in RL may massively hinder exploration, especially in non-linear environments, speaking of which --
- the paper only evaluate on classic mujoco, and lacks variety of evaluation task, speaking of evaluation --
- The evaluation also lacks other sota methods for comparison, since the promise was to have better performance, rather than better theoretical understanding, which goes back to my first point.

**Questions:**

How does sample rejection based on gradient relate to hessian?

---

> ### Author Response · Authors · 2024-11-24
> **Response to Reviewer xR4M**
>
> Thank you for your valuable comments.
> We will revise the paper based on your feedback.
>
> **Q1.**
> The paper do not have a theoretical underpinning for their approach though some reader may find the idea intuitive, that said, personally, I'm not a fan of removing samples from experience buffers;
>
> **A1.**
> We provide theoretical analyses in Appendix A to justify the use of PIToD.
>
>
> **Q2.**
> because the way I see it, deleting "negatively influential experiences" seems to be a lenient/hysteresis update commonly seen in optimistic approaches, it may be useful in same case but should be used with caution since it may hinder learning and cause bias; a comparison with similar approaches is not included. I think instead of removing samples, the safer approach is to use gradient clipping or learning rate schedules to stabilize training. Techniques like gradient clipping cap the maximum value of gradients to prevent instability without losing any data --
>
> **A2.**
> Our method (PIToD) is primarily designed to identify and remove experiences that hinder learning or introduce bias.
>
> Additionally, the main objective of our work is not to stabilize training but to estimate the influence of experiences.
> In the context of online reinforcement learning, our study is the first attempt to estimate the influence of experiences and, to the best of our knowledge, no similar studies (approaches) exist.
>
>
>
> **Q3.**
> Data removal in RL may massively hinder exploration, especially in non-linear environments, speaking of which --
>
> **A3.**
> In our experiments, data removal is not performed during training (c.f. Algorithm 4). Therefore, in our experiments, data removal does not hinder exploration.
>
>
> **Q4.**
> the paper only evaluate on classic mujoco, and lacks variety of evaluation task, speaking of evaluation --
>
> **A4.**
> In our paper, we conduct experiments on 4 Mujoco tasks and 8 DMC tasks (Section 6 and Appendix G).
> The number of tasks would be comparable to previous works accepted at ICLR (e.g., [1]).
>
> [1] https://openreview.net/forum?id=PczQtTsTIX
>
>
> **Q5.**
> The evaluation also lacks other sota methods for comparison, since the promise was to have better performance, rather than better theoretical understanding, which goes back to my first point.
>
> **A5.**
> We provide theoretical analyses (c.f. A1).
> Additionally, since our work is the first to address this type of study in online RL, there are no existing SOTA methods for comparison (c.f. A2).
>
>
> **Q6.**
> How does sample rejection based on gradient relate to hessian?
>
> **A6.**
> We may not fully understand your question. Are you asking whether our method (PIToD) relates to existing research (e.g., [2]), which uses Hessian-based approaches for influence estimation in a supervised learning setting?
>
> [2] https://arxiv.org/pdf/1703.04730

---

### Meta-Review · Area_Chair_2brp · 2024-12-09

**Metareview:**

This paper studies how to estimate the influence of a single entry in the experience replay. The naive leave-one-out approach is too expensive and this paper proposes a novel method based on masking the network parameters. That being said, reviewers are concerned about the theoretical justification of the proposed approach. In particular, it is not clear how removing experience would affect exploration of RL algorithms (e.g., Algorithm 3). To better justify the removal of experience, it is necessary to theoretically study how different experiences contribute to exploration and how the removal of experience affects credit assignment. Those effects cannot be characterized by the scalar metric used in the paper. Moreover, to demonstrate the usefulness of the proposed technique, it is worth considering more benchmark environments and more variants of using experience replays.

**Additional Comments On Reviewer Discussion:**

There is unfortunately not much discussion but reviewers are concerned about theoretical justification of the proposed approach and the significance of the results. The only positive reviewer is unable or unwilling to argue for acceptance.

---

### Decision · Program_Chairs · 2025-01-22

Reject